# KIF20A/MKLP2 regulates the division modes of neural progenitor cells during cortical development

Anqi Geng[1], Runxiang Qiu[1], Kiyohito Murai[1,6], Jiancheng Liu[1], Xiwei Wu[2], Heying Zhang[1], Henry Farhoodi[1], Nam Duong[1], Meisheng Jiang[3], Jiing-kuan Yee[4], Walter Tsark[5] & Qiang Lu[1]

Balanced symmetric and asymmetric divisions of neural progenitor cells (NPCs) are crucial for brain development, but the underlying mechanisms are not fully understood. Here we report that mitotic kinesin KIF20A/MKLP2 interacts with RGS3 and plays a crucial role in controlling the division modes of NPCs during cortical neurogenesis. Knockdown of KIF20A in NPCs causes dislocation of RGS3 from the intercellular bridge (ICB), impairs the function of Ephrin-B–RGS cell fate signaling complex, and leads to a transition from proliferative to differentiative divisions. Germline and inducible knockout of KIF20A causes a loss of progenitor cells and neurons and results in thinner cortex and ventriculomegaly. Interestingly, loss of function of KIF20A induces early cell cycle exit and precocious neuronal differentiation without causing substantial cytokinesis defect or apoptosis. Our results identify a RGS–KIF20A axis in the regulation of cell division and suggest a potential link of the ICB to regulation of cell fate determination.

[1] Department of Developmental and Stem Cell Biology, Beckman Research Institute of the City of Hope, Duarte, CA 91010, USA. [2] Department of Molecular and Cellular Biology, Beckman Research Institute of the City of Hope, Duarte, CA 91010, USA. [3] Department of Molecular and Medical Pharmacology, David Geffen School of Medicine, University of California, Los Angeles, Los Angeles, CA 90095, USA. [4] Department of Virology, Beckman Research Institute of the City of Hope, Duarte, CA 91010, USA. [5] Transgenic/Knockout Mice Facility, Beckman Research Institute of the City of Hope, Duarte, CA 91010, USA. [6] Present address: Department of Anatomy and Neurobiology, Nagasaki University School of Medicine, 1-12-4 Sakamoto, Nagasaki 852-8523, Japan. These authors contributed equally: Anqi Geng, Runxiang Qiu. Correspondence and requests for materials should be addressed to Q.L. (email: qlu@coh.org)

During brain development, neural progenitor cells (NPCs) have to maintain a tight control on the balance between proliferation and differentiation, so that desired neural cell types (including neurons, glia, and other cells) can be produced in an appropriate order and with the correct numbers. The regulation of such a fate decision in NPCs manifests in the form of symmetric (self-renewal) versus asymmetric (differentiation) cell divisions. Symmetric cell division expands the NPC pool, whereas asymmetric cell division allows NPCs to simultaneously maintain the progenitor pool and generate cellular diversity. The mechanisms that govern the mode of cell divisions (symmetric versus asymmetric) have been studied extensively in the nervous systems of *Drosophila* and *Caenorhabditis elegans*[1–4] and the knowledge gained from invertebrate studies has provided a framework to understand how symmetric and asymmetric cell divisions might be regulated in the mammalian systems. Consequently, homologs of many of the invertebrate genes implicated in regulation of symmetric versus asymmetric cell divisions were tested for a similar role as cell fate regulators in the mammalian brains. However, the mammalian studies have so far yielded contradicting results. For examples, knockout of Numb and Numbl was initially reported to compromise the maintenance of NPCs in the cortex[5,6], but it was later found to induce hyperproliferation of NPCs as well as impaired neuronal differentiation[7], or to cause delamination and displacement of apical radial glial cells (RGCs) into basal regions of the ventricular zone (VZ), but the progenitor fate was maintained[8]. Knockout of LGN/GPSM2, a modulator of G protein signaling, caused randomization of mitotic spindle orientation and ectopic distribution of the apical NPCs in the cortex, but did not have an obvious impact on proliferative versus neurogenic divisions[9,10]. Protein phosphatase PP4c was found to regulate spindle orientation and inhibit neurogenesis in one study[11] and was shown to promote neurogenesis and suppress NPC proliferation in another study[12]. These divergent results indicated that the actual process by which symmetric versus asymmetric cell division occurs in mammalian brains and the regulatory protein network of the process remain to be further explored[13,14]. The complexity of mammalian cells highlights the importance of identifying additional cell fate determinants crucial for regulating proliferative versus differentiative cell divisions, particularly those factors that work in close association with the cell division machinery.

We have previously shown that a regulator of G protein signaling (RGS) motif-mediated Ephrin-B reverse signaling pathway[15] and the Gα signaling pathway work together to regulate self-renewal and differentiation of NPCs during cortical neurogenesis[16–18]. RGS domain functions as a GTPase activating protein and links transmembrane receptor Ephrin-B to inhibition of Gαi and Gαo subunits. In the developing cortex, the Ephrin-B/RGS reverse signaling is required for the maintenance of NPCs[16,17], while Gα signaling functions to activate neurogenesis[18], and the balance between the two signaling pathways regulates the decision of NPCs to stay as a progenitor or to become a neuron[18]. In an earlier experiment, we found that in utero electroporation (IUE)-mediated knockdown of Ephrin-B1 or its cytoplasmic binding protein PDZ-RGS3 (RGS3 isoform 1) in the mouse cortex could induce early neuronal differentiation within 24 h of cell transfection[16]. This fast onset of knockdown effect prompted us to reason that the Ephrin-B/RGS signaling may be directly linked to cell division machinery, rather than modulating the outcome of NPC division via an indirect route, for example, through transcriptional regulation of downstream cell fate genes. We therefore searched for additional molecules that may interact with RGS3. In this study, we identify the interaction between KIF20A/MKLP2 and RGS3 within the intercellular bridge (ICB) of dividing NPCs. We further present functional data implicating

a key role of the RGS–KIF20A axis of interaction in cell fate determination during NPC divisions.

## Results

### KIF20A binds to the RGS domain of RGS3.
We screened an embryonic mouse yeast two-hybrid complementary DNA (cDNA) library using both the full-length RGS3 and a C-terminal fragment containing the RGS domain alone as bait. Full-length RGS3 identified a number of preys including α-tubulin and KIF20A (Supplementary Fig. 1) and the RGS domain pulled out KIF20A as the only prey. Because KIF20A was identified as a common candidate binding protein in both screens and it was known to be involved in cytokinesis[19], we characterized its potential interaction with RGS3 further. Mouse KIF20A is a mitotic kinesin with 887 amino acids. The two-hybrid screens pulled out a fragment of KIF20A from amino acids 625 to 818 (Supplementary Fig. 2a), which we called RGS-binding domain (RBD).

To characterize binding between RGS3 and KIF20A, constructs encoding myc-tagged RGS3 or derivatives and Flag-tagged KIF20A or derivatives were co-transfected into HEK293 cells. The two full-length proteins could co-immunoprecipitate (co-IP) from the transfected cells in reciprocal experiments (Supplementary Fig. 2b and 2c). Immunoprecipitation of RGS3 from embryonic brain lysate could bring down a protein band reactive to KIF20A antibody, suggesting that the two endogenous proteins were able to interact (Supplementary Fig. 2d). The RGS domain was essential for the interaction between RGS3 and KIF20A, because co-IP of the two proteins was impaired when the RGS domain was removed from RGS3 (Supplementary Fig. 2b and 2c). The importance of the RGS and RBD domains for interaction between RGS3 and KIF20A were further confirmed by reciprocal co-IP of the two domains with each other (Supplementary Fig. 2e and 2f). Furthermore, we found that KIF20A could interact with RGS domains of multiple subclasses in addition to that of RGS3 (Supplementary Fig. 2g).

### KIF20A and RGS3 co-localize in the intercellular bridge.
We next examined the subcellular localization of RGS3 and KIF20A in intact cells, using an inducible CHP100 cell line for expression of myc-tagged KIF20A based on a Tet-on system. The inducible system was used to help maintain a low level of expression of Myc-KIF20A so as to avoid cellular toxicity associated with overexpression of mitotic kinesins. Expression plasmids of RGS3 or derivatives were transfected into this cell line to assess co-localization with Myc-KIF20A. When Myc-KIF20A expression was induced, anti-myc staining could detect Myc-KIF20A in the mitotic cytoplasm of cells in early mitosis (Supplementary Fig. 3a). In telophase cells, Myc-KIF20A localization was prominent in the ICB/midbody (Supplementary Fig. 3b), consistent with previous studies[19–21]. In cells co-expressing GFP-RGS3 and Myc-KIF20A, green fluorescent protein (GFP) signal overlapped with the subcellular patterns of Myc-KIF20A in early mitotic cytoplasm (Supplementary Fig. 3c) and at the telophase (Supplementary Fig. 3d and 3e). This co-localization was dependent on binding between the two proteins, because the pattern of GFP signal was very different when a mutant GFP-RGS3 fusion protein lacking the RGS domain was co-expressed with Myc-KIF20A (Supplementary Fig. 3f and 3g). Gα subunit and Ephrin-B were known to interact with RGS3. To examine whether they could be localized in the ICB, expression plasmids of yellow fluorescent protein (YFP)-tagged Gα subunit (YFP-Gαi2) or hemagglutinin (HA)-tagged Ephrin-B1 (HA-Ephrin-B1) were transfected into CHP100 cells. YFP-Gαi2 was present in the midbody region of telophase cells (Supplementary Fig. 3h), consistent with a

previous report[22]. Similarly, HA-Ephrin-B1 could also be found in the midbody region overlapping with the localization of GFP-RGS3 when the two proteins were co-expressed in transfected cells (Supplementary Fig. 3i).

**KIF20A and RGS3 co-express in cortical NPCs.** We next examined expression of KIF20A in relation to RGS3 during embryonic cortical development. RNA in situ hybridizations showed that KIF20A transcript was specifically expressed in the

cortical ventricular and subventricular zones (VZ, SVZ) (Fig. 1a) where progenitor cells take residence, similar to the patterns previously observed for Ephrin-B1 and RGS3[15,16]. Anti-KIF20A staining showed that KIF20A protein was detectable in apically localized mitotic progenitor cells during cortical neurogenesis (Fig. 1b), consistent with KIF20A as a prominent mitotic protein. To facilitate examination of the subcellular position of KIF20A in the cortex, an expression plasmid of RFP-KIF20A fusion protein was first introduced into the cortex by IUE. We found that RFP-

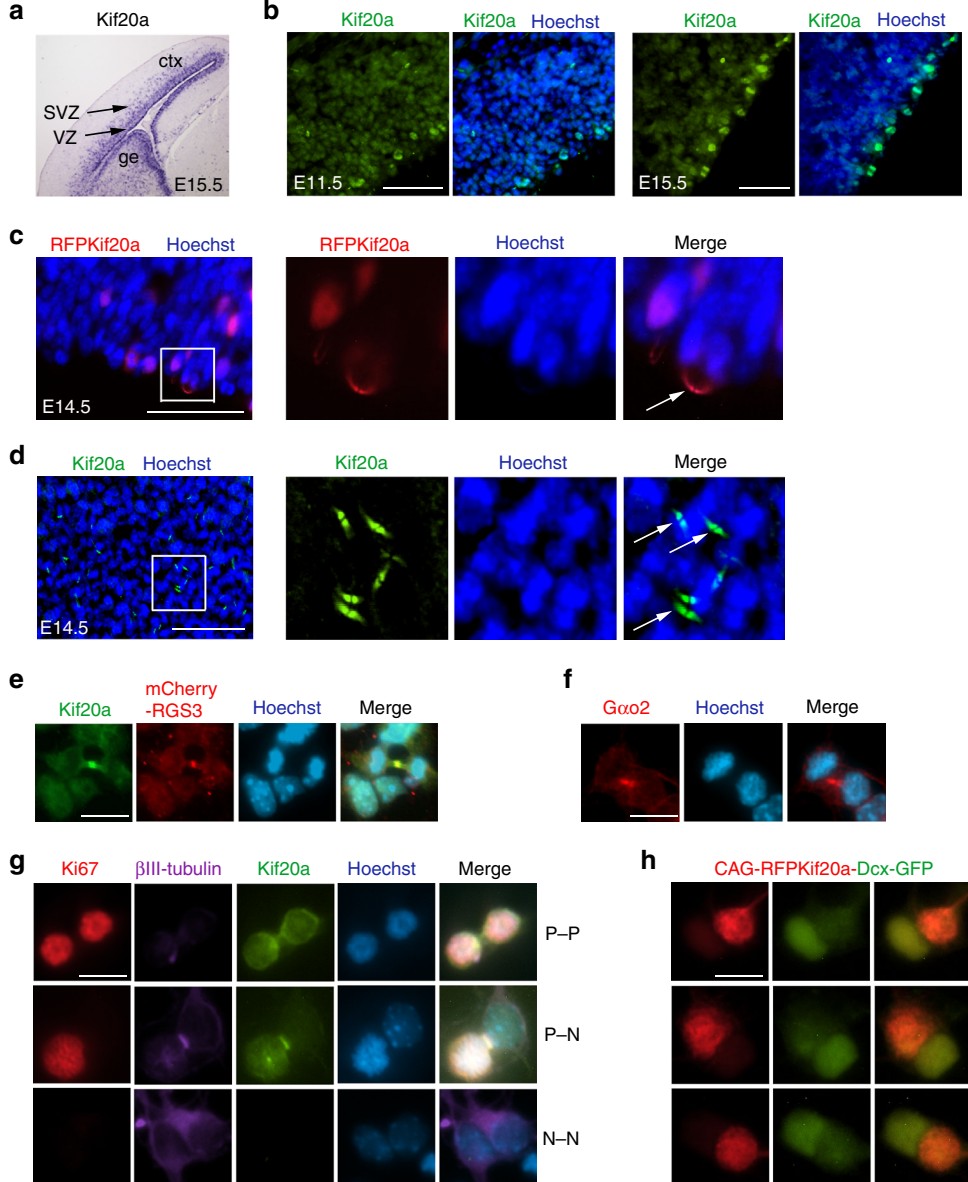

**Fig. 1** KIF20A, RGS3, and Gα subunit are present in the ICB of mitotic NPCs **a** RNA in situ hybridization revealed KIF20A mRNA in the germinal zone of the cortex where progenitor cells reside, similar to the transcripts of RGS3 and ephrin-B1 observed previously. ctx: cortex, ge: ganglionic eminence, SVZ: subventricular zone, VZ: ventricular zone. **b** Anti-KIF20A staining showed that KIF20A protein is more strongly expressed in apically localized mitotic cells. Scale bar represents 100 μm. **c** Expression of an RFP-KIF20A fusion protein in the cortex revealed its pattern in the presumptive intercellular bridge of apically dividing cells. Scale bar represents 100 μm. **d** En-face view of a whole-mount anti-KIF20A staining with E14.5 cortices revealed that KIF20A was present on the ventricular surface. Scale bar represents 100 μm. **e** In dissociated primary cortical progenitor cells with lentiviral expression of a mCherry-tagged RGS3 fusion protein, mCherry-RGS3 could be found in the midbody area of telophase cells with endogenous KIF20A. Scale bar represents 10 μm. **f** Immunostaining showed that Gαo2 could also be seen present at the midbody region of dividing cortical progenitor cells. Scale bar represents 10 μm. **g** Cells from the E14.5 mouse cortices were cultured for 20 h and stained for KIF20A, Ki67, and β-III-tubulin. Paired cells showing symmetric progenitor–progenitor self-renewal (P–P), asymmetric progenitor–neuron (P–N) differentiation, or symmetric neuron-neuron (N–N) differentiation were documented. KIF20A expression was selectively associated with daughter progenitor cells. Scale bar represents 10 μm. **h** Cortical cells electroporated with a plasmid carrying a CAG promoter-RFP-KIF20A cassette and a DCX promoter-GFP cassette were analyzed after 20 h in culture. In paired cells (showing three examples) with asymmetrical division (with daughter neurons marked by GFP), RFP-KIF20A, and GFP showed complementary patterns of expression. Scale bar represents 10 μm

KIF20A was present, facing the ventricle, within the presumptive intercellular bridge of apically dividing cells at the telophase (Fig. 1c). We next performed a whole-mount staining of the cortex and the resulting en-face view of the ventricular surface confirmed endogenous KIF20A expression in apically localized NPCs (Fig. 1d). To examine whether RGS3 could be present with KIF20A in the ICB of cortical NPCs, we introduced an expression plasmid of mCherry-tagged RGS3 (mCherry-RGS3) into cortical cells, because available RGS3 antibodies could not work well for detecting RGS3 in dividing cells. We found that mCherry-RGS3 and KIF20A showed overlapping patterns in the ICB of telophase NPCs (Fig. 1e). Multiple Gαi and Gαo subunits (RGS3 could target both Gαi and Gαo subfamilies) are expressed in cortical NPCs. Using an available antibody to Gαo2[23], we found Gαo2 subunit could also be seen in the ICB of dividing progenitor cells (Fig. 1f).

The post-mitotic midbody ring/remnant (MR), the center structure of the ICB, is either released from the dividing cells or inherited by one of the two daughter cells after membrane abscission[24]. It was reported that MR is associated with progenitor cell regulation[24,25]. We were therefore interested to assess whether KIF20A is present in post-mitotic MR. In cultured CHP100 cells, we found that structures marked by expression of KIF23/MKLP1, which were reminiscent of MR, did not show co-expression of KIF20A (Supplementary Fig. 4a). Similarly, within the cortical ventricular surface, KIF20A and MKLP1 could be seen co-expressed within the intact ICB but not in structures resembling post-mitotic MR (Supplementary Fig. 4b).

**KIF20A is selectively expressed in daughter NPCs.** Specific expression of KIF20A in mitotic progenitor cells of the cortex suggested that KIF20A might be important for maintaining a progenitor cell state. To explore this potential function, we first examined how KIF20A is expressed by daughter cells during NPC division using a paired cell assay[26]. Dissociated cells prepared from embryonic day 14.5 (E14.5) cortices were cultured for 20 h and then co-stained for KIF20A and cellular markers. Paired cells were documented for their immunostaining patterns. Our data (Fig. 1g) showed that KIF20A immunoreactivity is selectively associated with daughter progenitors (88%; $N = 222$), but rarely seen in daughter neurons (5%; $N = 343$). Alternatively, we constructed a CAG-RFP-KIF20A-Dcx-GFP plasmid which placed RFP-KIF20A fusion under the control of the CAG promoter and GFP under the control of the doublecortin (Dcx) promoter in the same plasmid. Expression of Dcx-driven GFP could mark the progression of a neuronal fate[27] and help visualize cell fate specification in daughter cells. We found that, among pairs of cells showing prominent GFP expression in one daughter cell (likely resulted from asymmetric mode of cell divisions), RFP-KIF20A and GFP displayed complementary expression levels (Fig. 1h), suggesting that KIF20A is linked to progenitor fate in daughter cells.

**Knockdown of KIF20A promotes differentiative NPC divisions.** We next generated plasmid-based small hairpin RNAs (shRNAs) against both the coding sequence (CDS) and the 3'-untranslated region (UTR) of KIF20A transcript (Supplementary Fig. 5). The shRNA plasmids were delivered into the mouse cortex at E13.5 by IUE and the transfected brains were analyzed at E15.5. We found that in brains electroporated with shKif20a-CDS or shKif20a-3'UTR, more transfected cells translocated from the VZ into the intermediate zone (IZ) and cortical plate (CP) compared with cells expressing a scrambled control shRNA (Fig. 2a), a phenomenon that reflects early neuronal differentiation and subsequent outward migration by the daughter neurons from the VZ to IZ and CP[16,18]. Consistent with activated neuronal differentiation, shKif20a-expressing cells that migrated out

earlier into the CP were positive for neuronal marker NeuN (Fig. 2b). The effect of shKif20a-3'UTR could be rescued by co-electroporating with the CDS of KIF20A (Fig. 2a), suggesting this was not due to an off-target effect. Cell cycle exit and re-entry analysis (Fig. 2c) further supported that knockdown of KIF20A induced more progenitor cells to leave the cell cycle. In addition, paired cell analysis using cells derived from the electroporated cortices showed that knocking down KIF20A expression caused more cells to adopt a differentiating fate (Fig. 2d), consistent with the idea of induced differentiation evidenced by cell distribution patterns seen in Fig. 2a.

To further examine the effect of KIF20A knockdown on cell fate decisions, we performed live cell imaging on cortical progenitor cells in active divisions. For this experiment, control and KIF20A shRNAs were introduced into cortical cells derived from the Dcx-mRFP transgenic reporter mice[27], so that emerging neuronal fate of daughter cells could be monitored with mRFP expression. Our results showed that knockdown of KIF20A expression caused a shift in progenitor cells from proliferative cell divisions to differentiative cell divisions (Fig. 3a; Supplementary Movies 1-6), consistent with data obtained from IUE-based assays. Lineage reconstruction from imaged divisions showed that cells expressing the control shRNA were often associated with more cycles of proliferative divisions, while cells expressing shKIF20A displayed more differentiative divisions (Fig. 3b–g). Knockdown of KIF20A in cortical cells did not cause obvious change of apoptosis status in the affected cortical cells (Supplementary Fig. 6). In addition, in all recorded divisions of progenitor cells expressing KIF20A-targeting shRNA, we did not see any obvious failures of cell abscission. These data thus collectively indicated that KIF20A is crucial for the maintenance of progenitor cell state and blocking KIF20A function results in early neuronal differentiation in cortical NPCs.

**KIF20A functions in coordination with the Ephrin-B/RGS complex.** Previously, we observed that co-expression of Ephrin-B1 and RGS3 could block differentiation, and as a result many transfected cells remained in the area of the subventricular zone (SVZ) and IZ, whereas cells expressing control GFP protein could progressively differentiate and migrate into the CP over 2–4-day period post electroporation[16] (Fig. 4). These cells showed expression of neural progenitor cell marker nestin but lacked the expression of neuronal marker βIII-tubulin[16], consistent with impaired neuronal differentiation. We used this phenomenon to ask whether KIF20A acts in association with the Ephrin-B/RGS function. When compared with brains electroporated with a scrambled control shRNA, shKif20a could reverse the inhibition of neuronal differentiation caused by over-expression of Ephrin-B1 and RGS3 (Fig. 4a). We next examined the effect of co-electroporation of shRNAs for KIF20A and RGS3 in the cortex. Our results showed that simultaneous knockdown of KIF20A and RGS3 caused neuronal differentiation and outward migration to the same extent of single knockdown of KIF20A (Fig. 2a), suggesting that KIF20A and Ephrin-B/RGS work in coordination in cell fate determination.

Because interaction with KIF20A was critical for the localization of RGS3 into the ICB in transfected cells (Supplementary Fig. 3e versus 3f), we further examined whether KIF20A could also facilitate localization of RGS3 into the ICB in dividing NPCs. As endogenous RGS3 was hard to detect in telophase NPCs due to poor sensitivity of the available antibodies, we introduced a mCherry-RGS3 fusion protein into cortical NPCs by lentiviral infection and expression. Lentivirus expressing either a scrambled shRNA or a specific shRNA targeting KIF20A was simultaneously introduced to cortical NPCs. Infected NPCs were cultured for

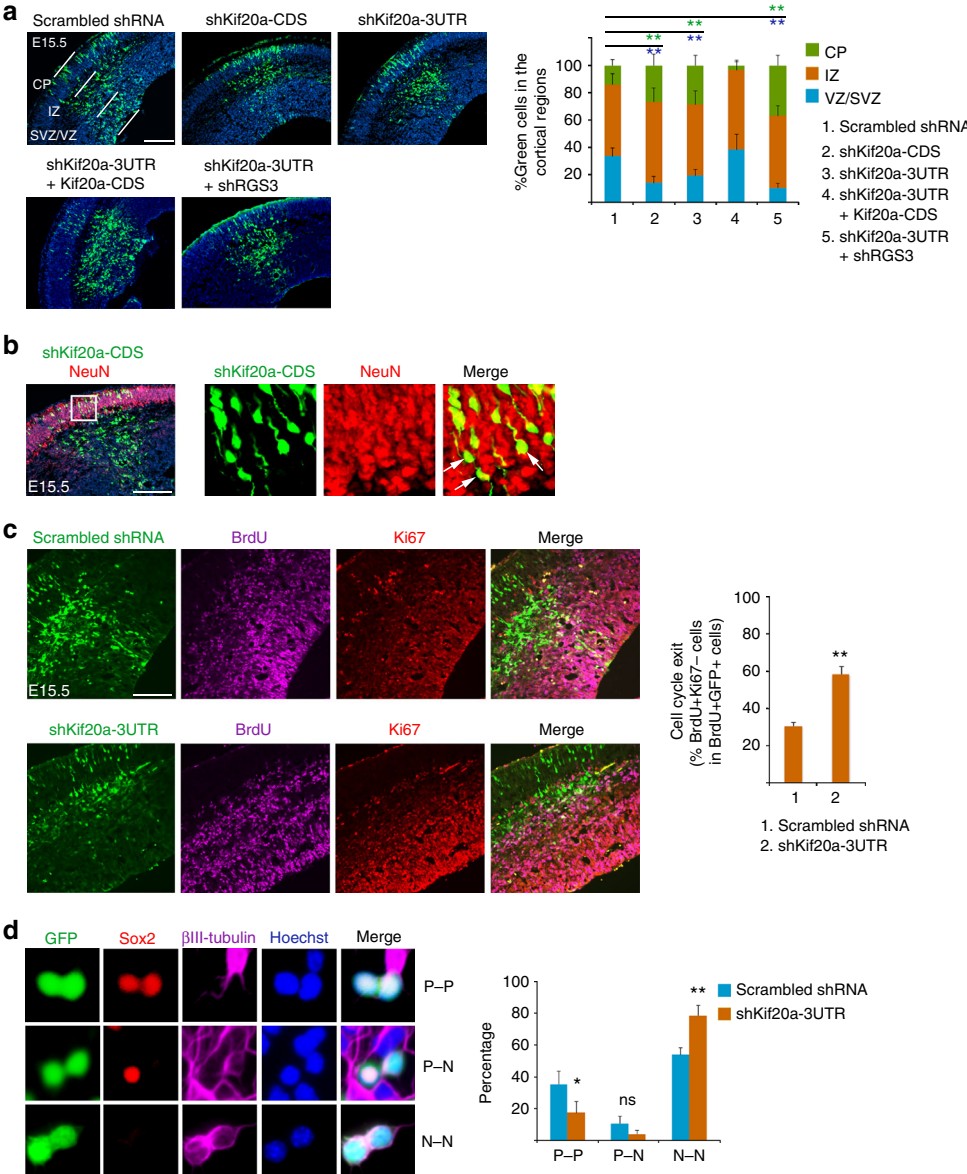

**Fig. 2** Knockdown of KIF20A causes neuronal differentiation in NPCs. **a** DNA plasmids were introduced into the cortex by IUE at E13.5 and the brains were analyzed at E15.5. The shRNA vector carried a ubiquitin promoter-GFP expression cassette for visualization of transfected cells. Distributions of transfected cells in different radial regions of the cortex were scored (5–8 electroporated brains for each plasmid or combination of plasmids were used for quantification). The symbol ** (in green and blue fonts) indicates $P < 0.01$ with respect to the CP and VZ/SVZ distributions compared to control shRNA. Scale bar represents 100 μm. Error bars represent SD. CP: cortical plate, IZ: intermediate zone, VZ/SVZ: ventricular/subventricular zone, CDS: coding sequence, 3'UTR: 3' untranslated region. **b** At 2 days after IUE-based electroporation, shKif20a-expressing cells (GFP$^+$ cells) in the CP were positive for NeuN, indicating that the cells were neuronal progeny. Arrows indicated examples of GFP$^+$NeuN$^+$ cells. Scale bar represents 100 μm. **c** DNA plasmids were introduced into the cortex by IUE at E13.5, BrdU was administered at E14.5, and the brains were collected at E15.5. Sections of electroporated brains were co-stained for incorporated BrdU and proliferating marker Ki67. Cell cycle exit index was calculated using the ratio of BrdU$^+$Ki67$^-$GFP$^+$ cells over the total population of BrdU$^+$GFP$^+$ cells. **$P < 0.01$ (Student's $t$-test). Scale bar represents 100 μm. **d** Dissociated cells from cortices electroporated with shKIF20A-3'UTR or control plasmid were cultured for 20 h and co-stained for Sox2 and β-III-tubulin. The percentage of paired cells showing P–P, P–N, and N–N divisions were scored (control shRNA, 451 cell pairs of three experiments; shKIF20A-3'UTR, 368 cell pairs of three experiments). Representative images of paired cells shown here were expressing shKIF20A-3'UTR. *$P < 0.05$, **$P < 0.01$ (Student's $t$-test); ns: not significant. Error bars represent SD

2 days and were then fixed and immunostained for KIF23/MKLP1, a marker of the midbody structure within the ICB. In control NPCs expressing a scrambled shRNA, mCherry-RGS3 was frequently seen associated with the midbody at telophase (Fig. 4b). However, when KIF20A expression was knocked down by KIF20A-targeting shRNA, fewer NPCs at telophase showed obvious accumulation of mCherry-RGS3 at the midbody region (Fig. 4b). These data together suggested that KIF20A and RGS3

function in a linear pathway in cortical NPCs to regulate the outcome of cell division and that the ICB is most likely the site where the RGS–KIF20A complex exerts their function on cell fate specification.

**Germline knockout of *Kif20a* leads to a defect in neurogenesis.**
To more conclusively understand the function of KIF20A in

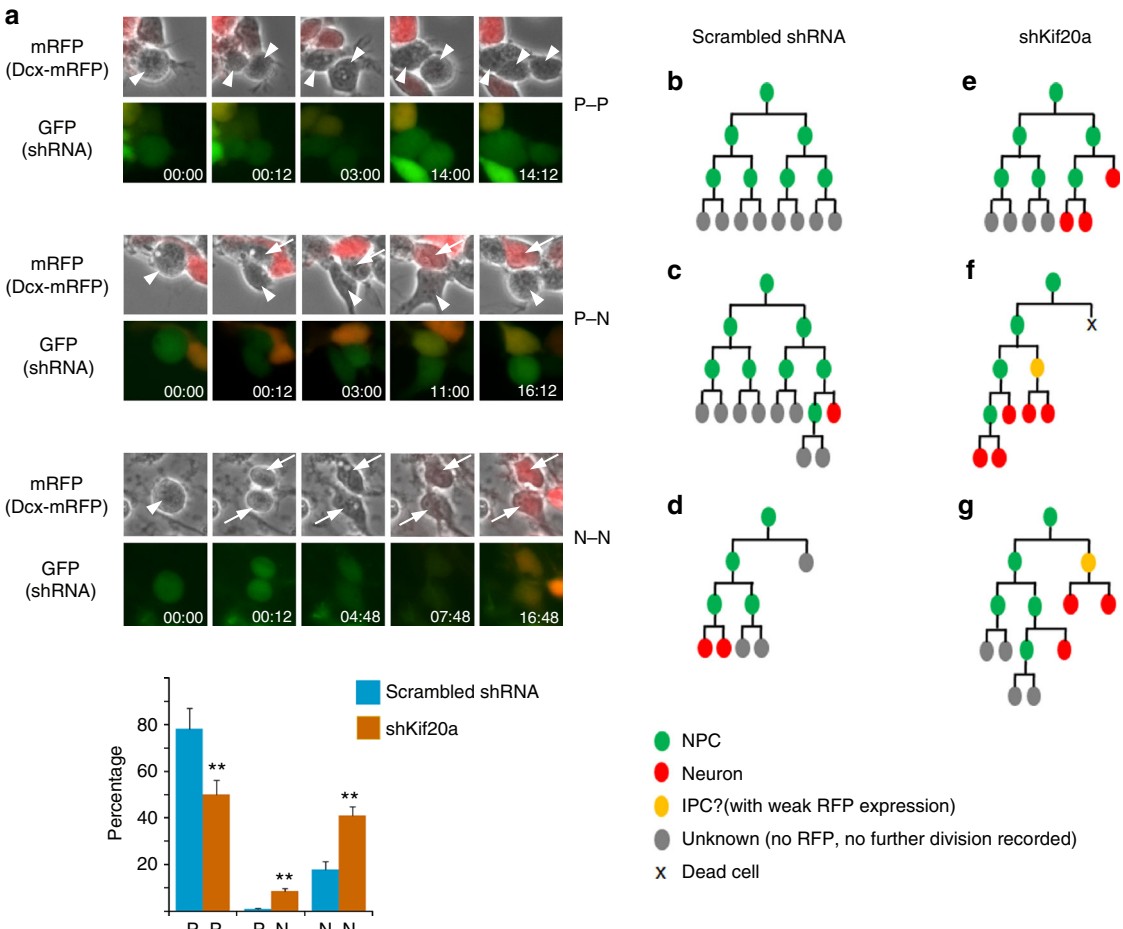

**Fig. 3** KIF20A is important for the balance between proliferative and differentiative cell divisions. **a** Dissociated cells from the E13.5 cortices of the Dcx-mRFP transgenic reporter mice were infected with lentiviruses for expression of shKif20a or scrambled control shRNA. After culturing for 1 day post infection, shRNA-expressing (GFP positive) progenitor cells were imaged in live culture at intervals of 12 min for 48–72 h. Phase contrast and fluorescent images at indicated time are shown. Appearance of RFP expression in some daughter cells reflected the emerging neuronal fate in these nascent cells. White arrowheads indicated mother and daughter progenitor cells. White arrows indicated daughter cells taking a neuronal fate. Progenitor–progenitor (P–P), progenitor–neuron (P–N), or terminal neuron–neuron (N–N) division modes were scored (in both shKif20a- and scrambled control shRNA-expressing progenitor cells, over 250 cell divisions were counted). Representative images of dividing cells shown here were expressing either the control shRNA (P–P) or shKIF20A-3'UTR (P–N and N–N). **P < 0.01 (Student's t-test). Error bars represent SD. **b**–**g** Representative lineage trees reconstructed from NPCs expressing scrambled control shRNA (**b**–**d**) or shKIF20A-3'UTR (**e**–**g**). NPC: neural progenitor cell, IPC: intermediate progenitor cell

cortical neurogenesis, we generated both germline and conditional *Kif20a* knockout mice (Supplementary Fig. 7). The homozygous *Kif20a* germline knockout mice displayed noticeable developmental abnormalities. At birth, no viable pups of homozygous *Kif20a* mutants were observed (Fig. 5a). At the mid-stage of cortical neurogenesis (E15.5), *Kif20a* mutant embryos were not recovered with the expected Mendelian ratio (Fig. 5a), indicating embryonic lethality. The surviving mutant embryos showed smaller body (not shown) and brain (Fig. 5b) sizes as well as reduced cortical thickness (Fig. 5c) compared to the wild-type littermates. Staining by βIII-tubulin antibody revealed that the *Kif20a* mutant brains had a thinner neuronal layer in the cortex compared to the same-stage littermates (Fig. 5d). Further examination of cellular markers of NPCs revealed that the *Kif20a* mutants had fewer Pax6+ RGCs and Tbr2+ intermediate progenitor cells (IPCs) compared to their wild-type littermates (Fig. 5e).

The loss of cortical NPCs in the *Kif20a* mutants could be a result from the following defects individually or in combination: a defect in NPC production, induced apoptosis, and/or premature differentiation. The first two possible abnormalities would not be

much unexpected as KIF20A was reported to be an important regulator of cytokinesis, the defect of which could impact cell proliferation and/or survival. The third possible abnormality was not obviously attributed to a regulator of cytokinesis, but could be inferred from our observed interaction between KIF20A and RGS3. To address these possibilities, we first examined whether loss of function (LOF) of KIF20A would result in cell death in the cortex. Detection of nicked DNA by TUNEL (terminal deoxynucleotidyl transferase dUTP nick end labeling) assay and staining of activated caspase 3 were performed for this purpose. At an earlier stage of cortical neurogenesis (E12.5), there was an increase in the number of cells undergoing apoptosis in the *Kif20a* mutant cortices compared to the wild-type littermates (Fig. 6a, b). As the neurogenesis progresses (E15.5), however, the mutant cortices showed no obvious difference in the level of apoptosis from the wild-type brains (Fig. 6a, b). We next examined whether there might be an increase of multinucleated cells in *Kif20a* mutant brains, which could be an indication of potential defect in cytokinesis. For this purpose, fluorescently labeled Concanavalin A (ConA) was used to help demarcate membrane peripherals of individual cells and nucleic acid stain

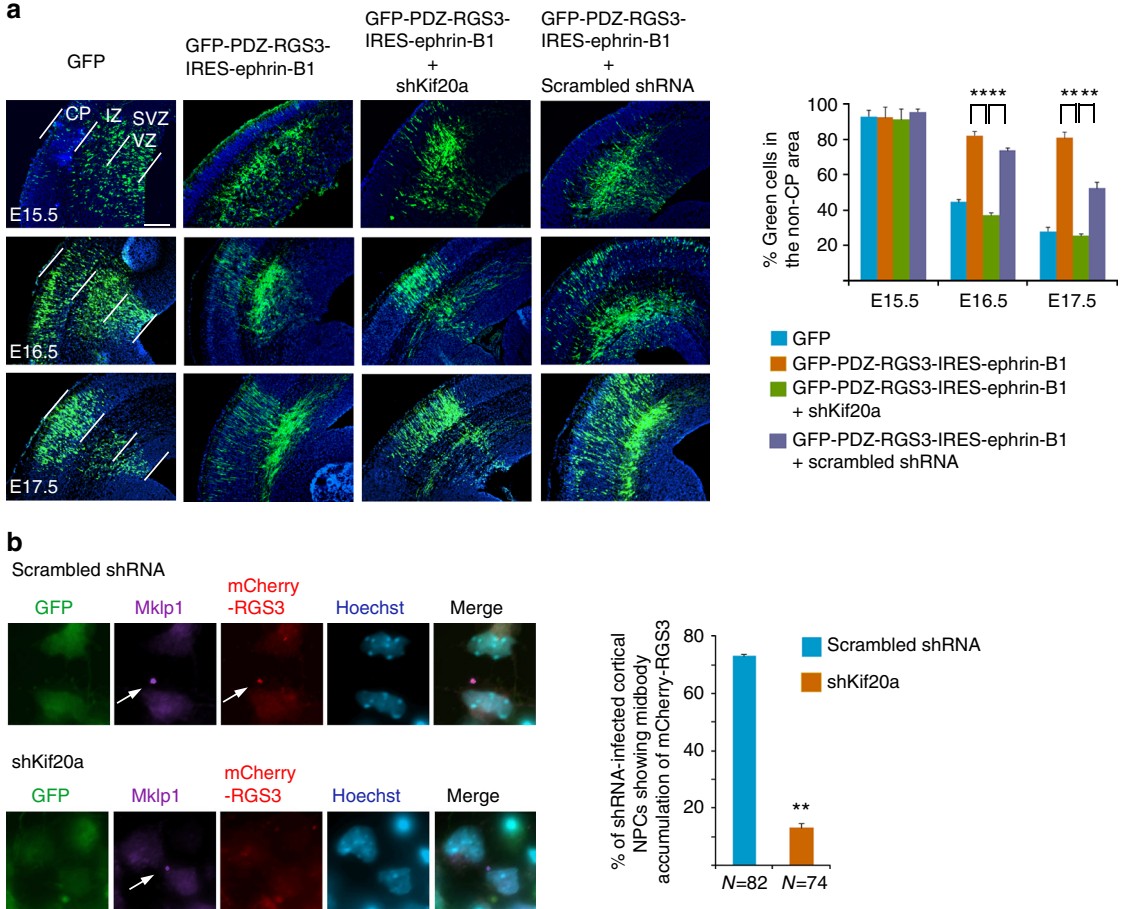

**Fig. 4** KIF20A works in coordination with the Ephrin-B–RGS complex in NPCs. **a** DNA plasmids were electroporated into the cortex at E13.5 and brains were analyzed at E15.5, E16.5, and E17.5. Distributions of transfected cells in the non-CP regions (IZ, SVZ, and VZ) were scored (5–8 electroporated brains for each plasmid or combination of plasmids were used for quantification). **P < 0.01 comparing rescue by shKif20a to controls. Error bars represent SD. Scale bar represents 100 μm. **b** Dissociated cells from the E13.5 cortices were co-infected with lentiviruses for expression of mCherry-RGS3/scrambled shRNA or mCherry-RGS3/shKif20a. After culturing for 2 days post infection, cells were fixed and stained with anti-KIF23/MKLP1 antibody. Percentage of telophase NPCs (with noticeable midbody localization of MKLP1, indicated by white arrows) showing midbody accumulation of mCherry-RGS3 was scored. **P < 0.01 (Student's t-test)

propidium iodide (PI) was used to mark nuclei (Fig. 6c). At both E12.5 and E15.5 stages, we found no obvious signs of multi-nucleated cells (more than one nucleus within ConA enclosed periphery) in the proliferating zone of the wild-type or mutant brains (E12.5 data shown in Fig. 6c). To better examine any possible defect of cytokinesis in a quantitative manner, we looked at cell cycle status of cortical NPCs in LOF of KIF20A. To facilitate recovering survival mutant embryos for analyses, we used a line of *Nestin-CreERT2*; *Kif20a^fl/fl* mice for inducible *Kif20a* knockout. *Nestin-CreERT2* transgene expression was shown to be efficiently activated by tamoxifen in neural stem/ progenitor cells[28], thus allowing inducible Cre-mediated deletion of floxed genes in NPCs. Tamoxifen was administered to pregnant mice at gestation days of E9.5 and E10.5 consecutively, and cortical cells were prepared from the littermate brains collected at E15.5, fixed, and stained with PI. Fluorescence-activated cell sorting (FACS)-based analysis showed that there was no difference in DNA content between the mutant (*Nestin-CreERT2*; *Kif20a^fl/fl*) and wild-type (*Kif20a^fl/fl*) cortical cells (Fig. 6d). Importantly, Cre-mediated inducible knockout of the *Kif20a* gene caused a similar defect of loss of cortical NPCs at E15.5 to what was observed in the E15.5 germline knockouts (Supplementary Fig. 8). These results together suggested that LOF of KIF20A did not cause obvious defect of cytokinesis in NPC

divisions or a possible defect of cytokinesis was largely compensated by other factors. Therefore, while the observed early increase in apoptosis could be a contributing factor to the overall loss of NPCs in the *Kif20a* mutant cortices, this factor alone did not appear to be able to fully account for the extent of the loss of cortical NPCs.

**Knockout of *Kif20a* leads to early cell cycle exit**. We next examined whether cell cycle exit and re-entry of cortical NPCs was abnormal in the *Kif20a* mutant brains. For this purpose, 5-bromo-2'-deoxyuridine (BrdU) was administered to pregnant female germline knockout mice at E14.5 and the brains of lit-termate embryos were collected for analyses 24 h later. At E15.5, a typical 24 h BrdU labeling yielded three ribbons of BrdU+ cells (Fig. 7a, wild-type (WT) panel). The most super-ficial ribbon (R1) represented cells that differentiated shortly after BrdU incorporation followed by outward migration of these newly born daughter neurons. The middle ribbon (R2) represented labeled cells that differentiated after going through an additional cell cycle. The apical ribbon (R3) contained labeled cells at mitosis and they might be poised for another wave of differentiation or re-entry to cell cycle. We noticed that the cortices of homozygous *Kif20a* mutant brains had an

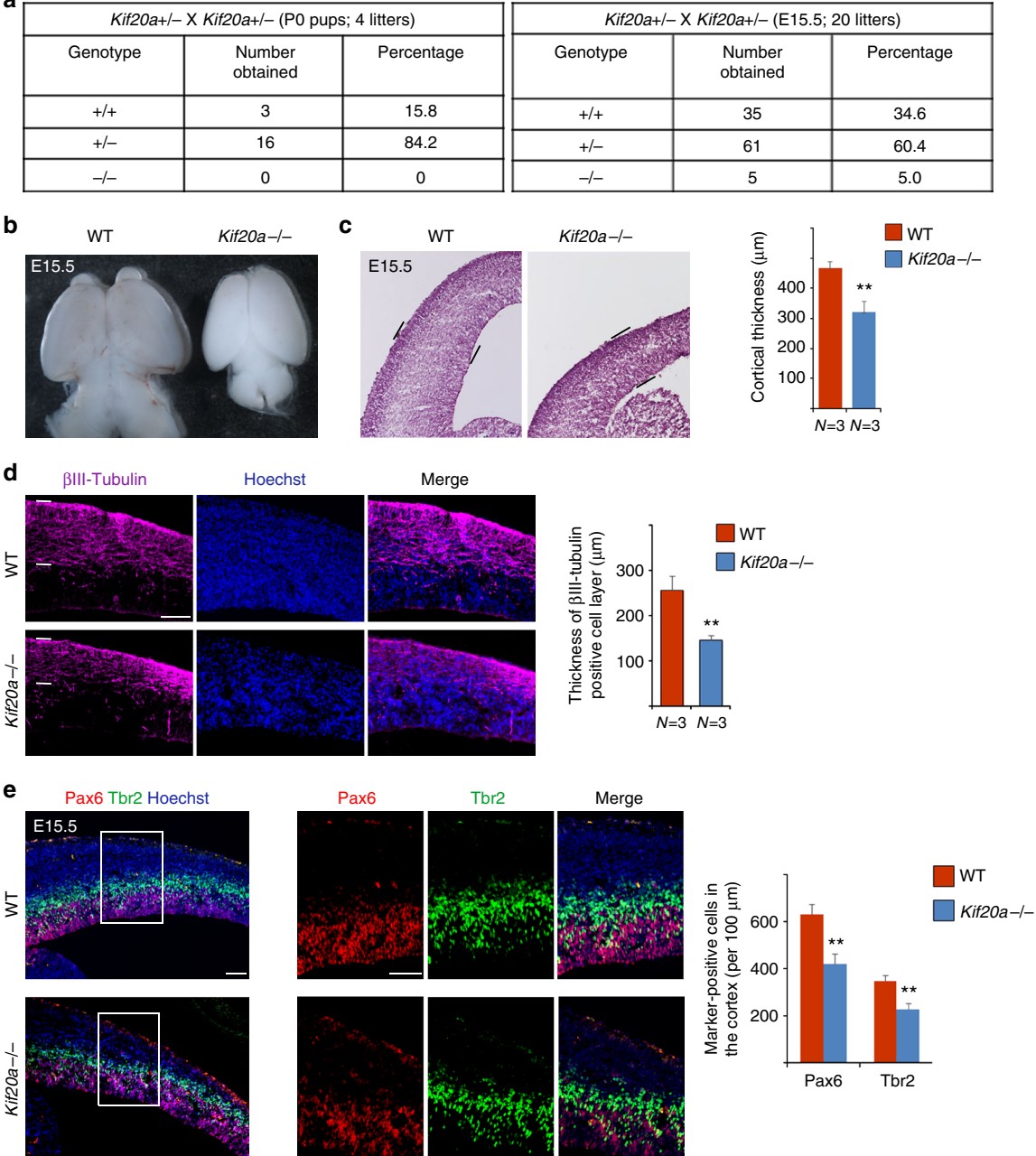

**Fig. 5** Germline knockout of *Kif20a* causes embryonic lethality and loss of cortical NPCs. **a** Low recovery rate of homozygous *Kif20a* knockout first pups or embryos showed embryonic lethality due to loss of function of KIF20A. **b** At the peak of cortical neurogenesis (E15.5), brains from the surviving *Kif20a* homozygous mutant embryos were smaller than their wild-type littermates. **c** Nissl staining of brain sections revealed thinner cortices of the *Kif20a* homozygous mutant brains at E15.5. **P < 0.01. **d** *Kif20a* homozygous mutant brains had fewer βIII-tubulin+ neurons at E15.5. **P < 0.01. Scale bar represents 100 μm. **e** *Kif20a* homozygous mutant brains had fewer Pax6+ radial glial cells and fewer Tbr2+ intermediate progenitor or basal progenitor cells at E15.5. **P < 0.01. Scale bar represents 100 μm

apparent thickening of the R2 ribbon but a significant loss of the R3 ribbon, as compared to the wild-type littermate brains (Fig. 7a). The thickening of R2 ribbon in the mutant cortices suggested that during this cell cycle, more labeled cells might have differentiated than those in the wild-type cortices. The loss of R3 ribbon was likely a cumulative result caused by early depletion of the NPC pool during previous cycles of cell divisions. Quantification revealed that there was a higher portion of BrdU+Ki67− cells (cells exited the cell cycle) in the BrdU+ population of the mutant cortices than that of the wild-type littermate cortices (Fig. 7a), suggesting an early cell cycle exit of

the mutant NPCs. Further examination of cell marker expression showed that BrdU+Ki67− cells that had migrated into the CP were positive for neuronal marker NeuN (Fig. 7b), suggesting that mutant NPCs, which left the cell cycle earlier, had differentiated into neurons.

To further evaluate the cell fate function of KIF20A in cortical neurogenesis, we next examined *Kif20a* conditional knockout mice. Because knockdown of KIF20A by specific shRNAs in the cortex caused precocious neuronal differentiation, we reasoned that expression of a Cre enzyme in the cortices of the *Kif20a^{fl/fl}* homozygous mice would be expected to induce a similar

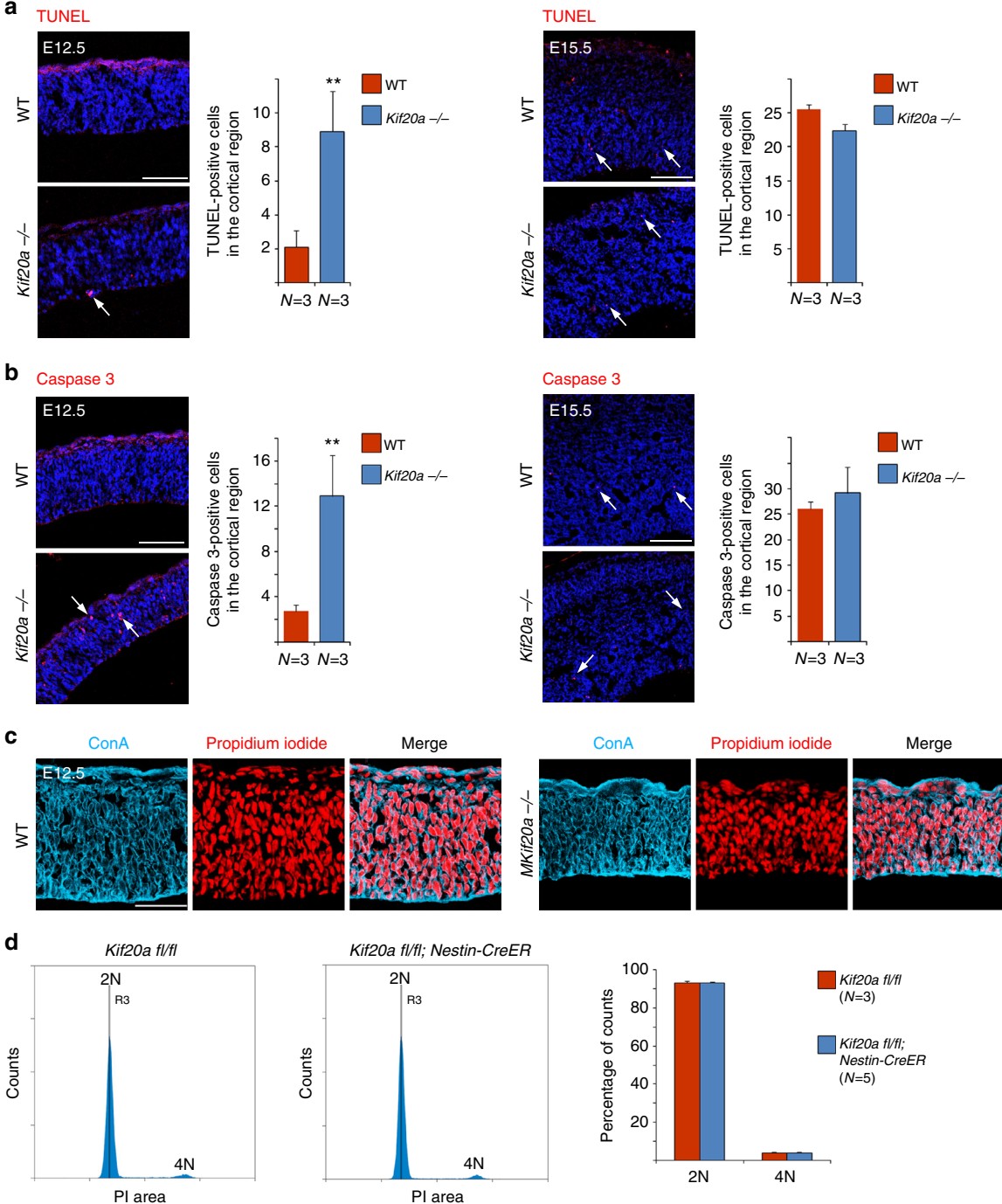

**Fig. 6** Knockout of *Kif20a* causes a subtle increase in apoptosis without obvious defect in cytokinesis. **a** TUNEL staining for visualizing cells in apoptosis. An increase of numbers of apoptotic cells in the *Kif20a* germline knockout mutant cortices was seen at E12.5. Apoptosis status in the mutant and wild-type cortices appeared comparable at E15.5. Scale bar represents 100 μm. TUNEL-positive cells in the entire cortical region were counted. \*\**P* < 0.01. **b** Anti-activated caspase 3 staining for detecting cells in apoptosis. Staining of activated caspase 3 showed a similar trend of apoptosis status in the germline knockout mutant and wild-type cortices to TUNEL staining. Scale bar represents 100 μm. Caspase 3-positive cells in the entire cortical region were counted. \*\**P* < 0.01. **c** No obvious bi- or multinucleated cells were seen in the wild-type or *Kif20a* germline knockout mutant cortices. ConA (cyan) and propidium iodide (red) were used for visualizing cell peripherals and nuclei, respectively. Scale bar represents 100 μm. **d** Cell cycle analysis of wild-type and mutant cortical cells. Tamoxifen was administered to *Kif20a* inducible knockout mice at E9.5 and E10.5 consecutively. Littermates of *Nestin-CreERT2*; *KIF20A^fl/fl* and control *KIF20A^fl/fl* embryos were collected at E15.5. Cortical cells were examined for their DNA contents by FACS analysis. PI propidium iodide, 2N cells in G1/S phase, 4N cells in G2/M phase or bi-nucleated cells. No obvious difference in DNA content of cortical cells was observed (*P* = 0.73 and 0.93 for 2N and 4N, respectively)

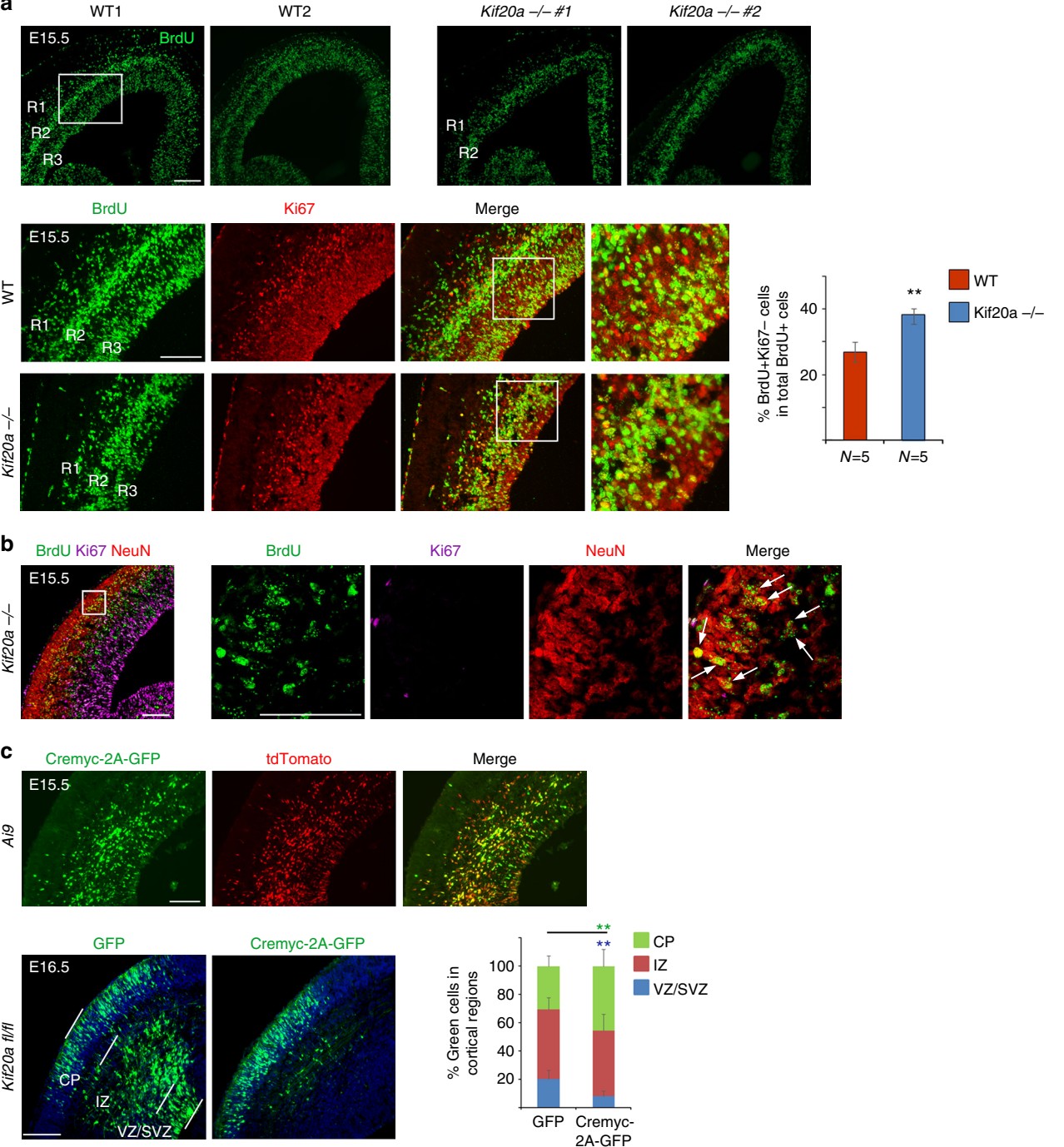

**Fig. 7** Knockout of *Kif20a* causes early cell cycle exit and precocious neuronal differentiation. **a** BrdU was given to pregnant mouse at E14.5 and embryonic brains were collected 24 h afterwards. R1, R2, and R3 indicated the BrdU-labeled cell ribbons located progressively from superficial regions to apical surface. BrdU$^+$Ki67$^-$ cells were progenitor cells that had left the cell cycle. More BrdU$^+$Ki67$^-$ cells in the total population of BrdU$^+$ cells were present in the cortices of the *Kif20a* knockout-first embryos. Scale bar represents 100 μm. **P < 0.01. Error bars represent SD. **b** BrdU$^+$Ki67$^-$ cells that had migrated into the cortical plate in the *Kif20a* knockout-first cortex were positive for NeuN, indicating that these cells had become neurons. Arrows indicated examples of BrdU$^+$Ki67$^-$NeuN$^+$ cells. Scale bar represents 100 μm. **c** In the upper panel, CAG-Cremyc-2A-GFP plasmid was introduced into the cortices of Ai9 (loxP-STOP-loxP-tdTomato) reporter mice at E13.5 by IUE. Electroporated brains were harvested at E15.5 for analysis. Majority of Cre-expressing cells (GFP$^+$ cells) were positive for tdTomato expression, suggesting that Cre-mediated recombination was successfully carried out in these cells. In the lower panel, CAG-Cremyc-2A-GFP or control GFP plasmids were introduced into the cortices of *Kif20a$^{fl/fl}$* conditional knockout mice by IUE at E13.5 and the brains were analyzed at E16.5. Distributions of transfected cells in different radial regions of the cortex were scored. Scale bar represents 100 μm. The symbol ** (in green and blue font) indicates P < 0.01 with respect to the CP and VZ/SVZ distributions comparing to control GFP. Error bars represent SD. CP: cortical plate, IZ: intermediate zone, VZ/SVZ: ventricular/subventricular zone. Scale bar represents 100 μm

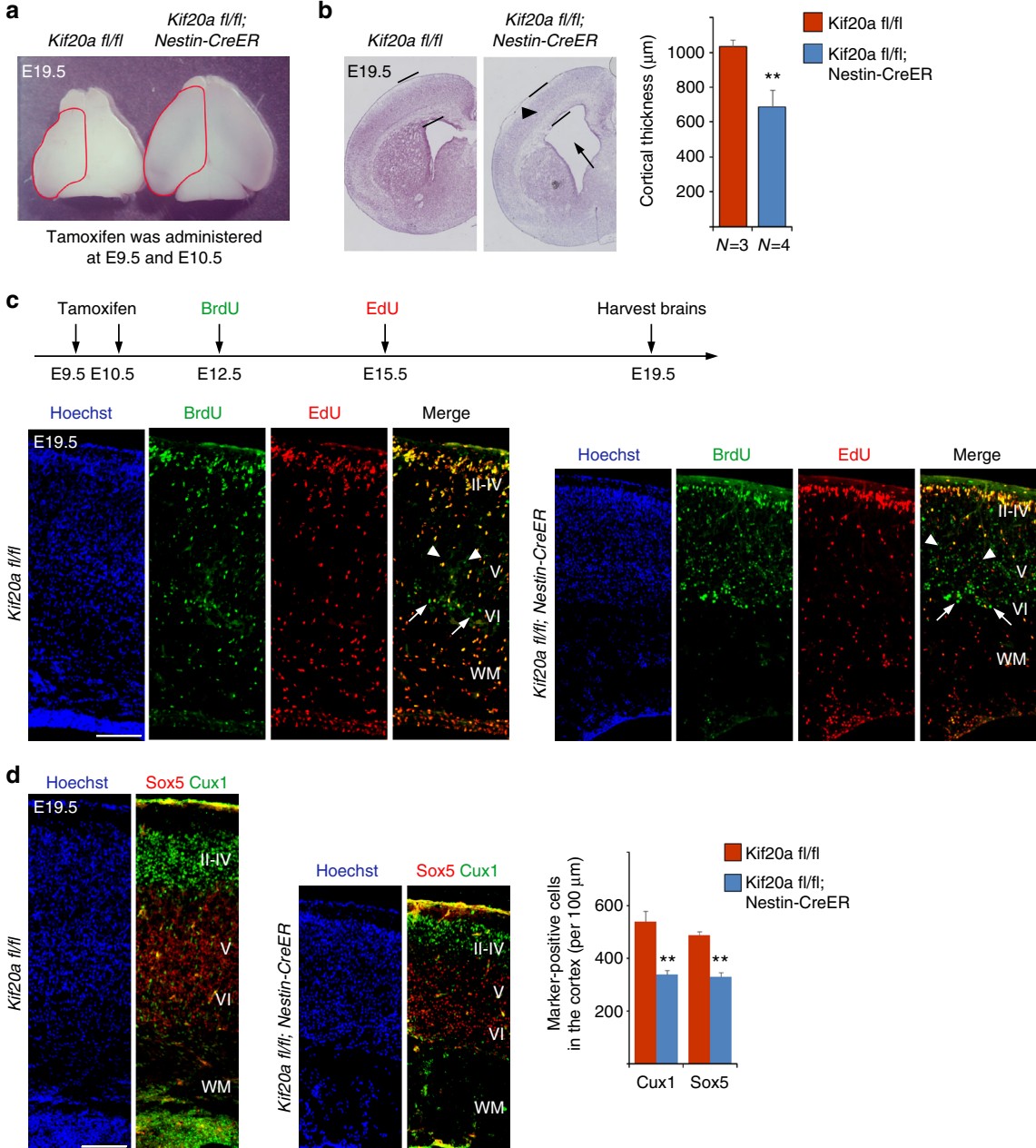

**Fig. 8** Inducible knockout of *Kif20a* in NPCs causes ventriculomegaly and defect in cortical neuronal production. **a** Tamoxifen was administered to pregnant mice carrying *Nestin-CreERT2; KIF20A*$^{fl/fl}$ and control *KIF20A*$^{fl/fl}$ littermates at E9.5 and E10.5. Pups were delivered by C-section (E19.5). Inducible knockout of *Kif20a* yielded brains with elongated cortical hemispheres. **b** Nissl staining of brain sections revealed thinner cortices (black arrowhead) and dilated ventricles (black arrow) of the *Kif20a* mutant brains. **P < 0.01. Error bars represent SD. **c** BrdU and EdU were sequentially administered at E12.5 and E15.5, respectively. In both *Nestin-CreERT2; KIF20A*$^{fl/fl}$ mice and control *KIF20A*$^{fl/fl}$ mice, BrdU$^+$EdU$^-$ cells (early-born neurons) and BrdU$^+$EdU$^+$ cells (late-born neurons) were located in the respective deep and upper layers, suggesting *Kif20a* knockout did not affect cortical lamination. More BrdU$^+$EdU$^-$ cells could be seen in the deep layers of *Kif20a* mutant brains, reflecting early cell cycle exit due to LOF of KIF20A. Arrows and arrowheads indicated bright and dim BrdU positive cells, respectively. **d** SOX5$^+$ and CUX1$^+$ cortical neurons in both *Nestin-CreERT2; KIF20A*$^{fl/fl}$ mice and control *KIF20A*$^{fl/fl}$ mice were seen correctly located in deep layers and upper layers, respectively. Neuronal productions, however, were compromised in the cortices of the mutant mice. Scale bar represents 100 μm. **P < 0.01. Error bars represent SD

phenotype. To perform this experiment, we prepared a plasmid CAG-Cremyc-2A-GFP for co-expression of Cre enzyme and GFP under the control of CAG promoter. When electroporated into the cortices of the Ai9 tdTomato reporter mice, this plasmid could lead to Cre-mediated recombination in majority of transfected cells, evident by co-expression of GFP with tdTomato (Fig. 7c, *Ai9* panel). To examine the effect of conditional deletion of the *Kif20a* gene, the CAG-Cremyc-2A-GFP plasmid or a control expression plasmid of CAG-GFP was electroporated into

the cortices of *Kif20a*$^{fl/fl}$ embryos at E13.5. The transfected brains were then collected for analyses at E16.5. Cre-expression caused more cortical cells to translocate into the IZ and CP than expression of the control GFP alone (Fig. 7c, *Kif20a*$^{fl/fl}$ panel), reflecting early differentiation and subsequent outward migration of the transfected cells. Consistent with the idea of activated neuronal differentiation (rather than abnormal NPC displacement), Cre-expressing cells that migrated out earlier into the CP were positive for neuronal marker NeuN (Supplementary Fig. 9a).

The induced differentiation due to LOF of KIF20A was further supported by an acute cell culture quantification assay (Supplementary Fig. 9b), in which more Cre-expressing cells derived from electroporated cortices were shown positive for βIII-tubulin compared to GFP-expressing control cells, and fewer of these cells expressed proliferating cell marker Ki67.

**Inducible *Kif20a* knockout leads to a defect in neurogenesis**. We next used inducible deletion of the *Kif20a* gene in NPCs to further examine the function of KIF20A in the progression of cortical morphogenesis. Tamoxifen was administered to pregnant mice at gestation days of E9.5 and E10.5 consecutively, and the littermate brains were collected for analyses around birth (E19.5) by Cesarean (C)-section delivery of pups. To facilitate tracking of early-born and late-born neurons, BrdU and 5-ethynyl-2'-deoxyuridine (EdU) were administered to pregnant mice sequentially at E12.5 and E15.5, respectively. At E19.5, the *Nestin-CreERT2*; *Kif20a^fl/fl^* brains appeared to show elongated cortical hemispheres compared to their control *Kif20a^fl/fl^* littermate brains (Fig. 8a). Sections of these brain samples revealed that the *Nestin-CreERT2*; *Kif20a^fl/fl^* brains had thinner cortices and wider ventricles than the *Kif20a^fl/fl^* brains (Fig. 8b), suggesting that elongation of the mutant cortices was due to enlargement of the lateral ventricles. Compared to the control *Kif20a^fl/fl^* brains, the white matter area in the *Nestin-CreERT2*; *Kif20a^fl/fl^* brains had significantly fewer cells present and also displayed apparent tissue disintegration (Fig. 8c, d). Co-staining of BrdU and EdU showed that BrdU$^+$EdU$^-$ cells (early-born neurons) and BrdU$^+$EdU$^+$ cells (late-born neurons) in both the *Nestin-CreERT2*; *Kif20a^fl/fl^* and control *Kif20a^fl/fl^* brains were located in respective deep and upper cortical radial domain (Fig. 8c), suggesting that cortical lamination in the mutant brains appeared to be normal. This was further supported by the expression patterns of SOX5 (early-born neuron marker, deep layer) and CUX1 (late-born neuron marker, upper layer) (Fig. 8d). A closer examination of BrdU- and EdU-labeled cells revealed that there were at least two groups of BrdU$^+$EdU$^-$ cells present in the deep layer vicinity, one group with brighter fluorescent intensity of BrdU and the other with reduced intensity (the latter group likely represented differentiated cells that had diluted its BrdU content due to additional cycles of cell divisions). Noticeably, more BrdU$^+$EdU$^-$ cells (particularly the group of cells with lighter BrdU staining intensity) were present in the deep layer region of *Nestin-CreERT2*; *Kif20a^fl/fl^* brains than in the *Kif20a^fl/fl^* control brains (Fig. 8c), reflecting early cell cycle exit and precocious neuronal differentiation occurred in the mutant brains. Furthermore, in the *Nestin-CreERT2*; *Kif20a^fl/fl^* brains, there were fewer early-born neurons (SOX5$^+$) and late-born neurons (CUX1$^+$) compared to the *Kif20a^fl/fl^* control brains, with the late-born neuron numbers more significantly reduced (Fig. 8d). These data suggested that LOF of KIF20A could tip the balance between self-renewal and differentiation in NPCs toward differentiation, leading to a depletion of NPCs and resulting in a severe impairment of cortical neuron production.

**LOF of KIF20A leads to changes of gene expression patterns**. Using both the germline and inducible *Kif20a* knockout mice, we collected littermates of the E12.5 wild-type and mutant cortices to examine gene expression status of cortical cells due to inactivation of the *Kif20a* gene (for inducible knockouts, tamoxifen was given at E9.5 and E10.5 consecutively). RNA-sequencing (RNA-seq) data comparing wild-type and mutant brains identified 80 down-regulated genes and 193 up-regulated in the germline mutant brains and 59 down-regulated genes and 47 up-regulated in the inducible knockout brains (fold change ≥ 1.5; *P* value ≤ 0.05)

(Supplementary Fig. 10a and 10b; Supplementary Data 1 and 2). DAVID (the database for annotation, visualization and integrated discovery) bioinformatics analyses (DAVID 6.8) of the differentially expressed genes from the germline knockouts showed that the top biological processes in down-regulated genes highlighted genes with roles in regulation of cell cycle and cell death (Supplementary Fig. 10c), whereas those in up-regulated genes revealed enrichment of genes involved in morphogenesis and cell differentiation. DAVID analyses of the differentially expressed genes from the inducible knockouts were less obvious with respect to the trends of gene expression functions (Supplementary Fig. 10c), perhaps due to the heterogeneity of varied gene deletion efficiency in mutant cells. While further pathway analyses of the differentially expressed genes from the germline knockouts did not consistently reveal any particular signaling pathways specifically affected by LOF of KIF20A, down-regulation of cell cycle-related genes and up-regulation of differentiation-related genes in the mutant cortex appeared to be overall consistent with the switch of NPCs from a proliferation state to a differentiation state as a result of the *Kif20a* knockout.

## Discussion

Our data demonstrate that KIF20A is essential for maintaining progenitor cell state and acts, at least in part, by facilitating the localization of RGS3 to the ICB/midbody of dividing NPCs where KIF20A, RGS3, and Gα subunit may work together to control the outcome of cell division. Symmetric versus asymmetric cell division is thought to be achieved through dividing progenitors, giving daughter cells different sets or amounts of cell fate determinants[1–4]. However, the mechanism by which cell fate determinants are differentially segregated is not well understood. Our data suggest that daughter cell asymmetry may arise during the late stage of cytokinesis in association with the ICB and the function of KIF20A. We speculate that, being a kinesin motor protein, KIF20A may act to regulate cell fate by facilitating symmetric or asymmetric delivery of cargos into the daughter cells. Such cargos are likely in the form of vesicles, protein/protein complexes, and/or protein/RNA complexes, which are expected to contain various cell fate determinants. KIF20A, and likely some other mitotic kinesins, may be critical for the trafficking of such cell fate determinants between the two daughter cells, which may affect the distribution or dynamics (stability) of these proteins and/or RNAs in the two nascent cells resulting in symmetric or asymmetric outcome of cell division (e.g., in the form of eventual proliferative or differentiative state of transcriptomes).

In our analyses of germline *Kif20a* knockout mice, we observed that LOF of KIF20A caused a more severe abnormality in embryonic development than that displayed in the RGS3 knockout mice[17]. This stronger phenotype of the *Kif20a* knockout mice was likely due to the added effect of cytokinesis defect which could lead to apoptosis of mutant NPCs in the early stage of cortical neurogenesis. On the other hand, our observations also revealed that KIF20A was an important regulator in cell fate determination. This raised an interesting question of whether these two functions of KIF20A were interconnected by a common mechanism or mediated through distinct actions. The fact that, in the absence of KIF20A, many NPCs completed cell division to give rise to new neurons that could migrate normally out of the VZ indicated that KIF20A might contribute to cytokinesis and cell fate determination via independent mechanisms. In the mutant NPCs, the cytokinesis function of KIF20A appeared to be compensated to a large extent by other mechanisms, either by a remaining homologous protein such as KIF20B[29] or by other redundant mechanisms, whereas the cell fate function of KIF20A was more severely compromised. On

the basis of this reasoning, we speculate that KIF20A may play two important and sequential roles during the division of NPCs (Fig. 9): (1) the canonical role for supporting the ordered progression of cytokinesis (e.g., by targeting Aurora B to midbody)[30,31] during midbody assembly; and (2) a role for controlling specification of the fate of daughter cells likely at the stage of cell abscission. KIF20A was found to interact with RGS3 (this study) and DLG5[32], the two regulators that are crucial for cortical neurogenesis[17,33]. These specific protein–protein interactions may aid in the recruitment or targeting of KIF20A cargos within the ICB. It is also conceivable that KIF20A may achieve its cell fate function by coordinating with the abscission machinery, including membrane/vesicle trafficking proteins[34] and the Endosomal Sorting Complexes Required for Transport (ESCRT)[35,36]. While the precise mechanisms by which these factors residing in the ICB may control proliferative versus differentiative divisions await further investigation, this study indicates that the ICB is a potential candidate site where cell fate regulators work coordinately, leading to maintenance of symmetry or generation of asymmetry within two nascent daughter cells. In addition, as cortical NPCs divide along the apical surface of the ventricular zone, the observation that the ICB (and therefore the important regulators involved in cell abscission) orients towards the ventricle is intriguing in view of the discovery of the importance of cerebrospinal fluid (CSF) in guiding neurogenesis in the mammalian brains[37–39]. Neighboring cells, without direct contact with the dividing progenitors, may be able to influence the outcome of cell division via secreted factors into the CSF, allowing both stochastic and regulated NPC cell divisions.

KIF20A is ubiquitously expressed in all proliferating cells, particularly associated with the mitotic state of dividing cells, and should be expected to be present in stem/progenitor cells of various tissues. This raises a question of whether the cell fate function of KIF20A that we observed for cortical NPCs in this study may be a common mechanism shared by other stem/progenitor cell systems. Within the central nervous system (CNS), besides the embryonic cerebral cortex, Ephrin-Bs and RGS3 were expressed in most, if not all, other types of stem/progenitor cells, including those of the embryonic cerebellum, spinal cord, and adult SVZ[15,40–42] (and our unpublished observations). It is therefore conceivable that the RGS3–KIF20A interaction also plays a crucial role in NPCs of these different CNS regions to regulate cell fate specification. Outside the CNS, we speculate that an RGS–KIF20A interaction may function similarly in stem/progenitor cells of various other tissue types to control the balance between self-renewal versus differentiation. Because RGS proteins have emerged as a large family of proteins with more than 30 members[43], it is likely that the specificity of this function may reside in the RGS proteins. For instance, different tissue-specific stem/progenitor cell systems may harbor distinct members of the RGS proteins that can work with KIF20A. In this regard, we observed that KIF20A could interact with several other subclasses of RGS proteins in addition to RGS3 (Supplementary Fig. 2g), making possible distinct RGS–KIF20A interaction in different stem/progenitor cell types. The potential function of RGS–KIF20A interaction in different stem/progenitor cell systems will certainly require further investigation.

Two previous studies reported a potential link of midbody remnant (MR) to progenitor cell fate determination. However, one study suggested that the inheritance of MR was associated with progenitor identity[24], whereas the other study indicated that the release of MR was a characteristic feature of progenitor cells[25]. Later studies with time-lapse imaging of zygote division in *C. elegans*[44] or germline stem cell division in *Drosophila*[45] revealed that MR, marked by expression of ZEN-4/PAVAROTTI (homologs of mammalian KIF23/MKLP1), could be stereotypically inherited by either daughter progenitor or differentiating

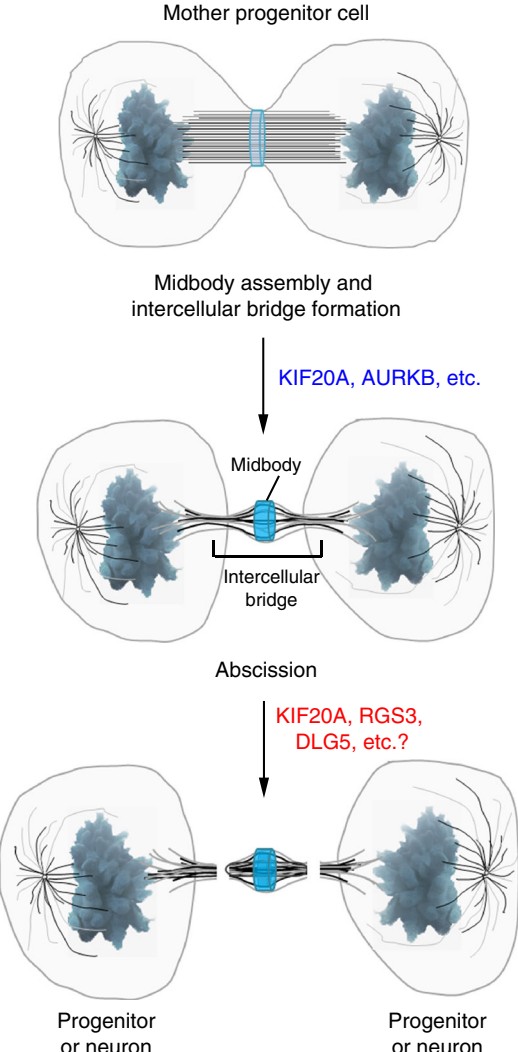

**Fig. 9** A working hypothesis for two sequential functions of KIF20A during NPC divisions. In its better known function discovered in previous studies[30, 31], KIF20A is involved in the assembly of the midbody, for example, facilitating translocation of Aurora B (AURKB) into the midbody. This function of KIF20A is important for an ordered progression of cytokinesis. In addition, data obtained from the current study suggest that KIF20A also plays a role in cell fate determination, being essential for the maintenance of the proliferative state of NPCs. This latter function of KIF20A likely occurs during the stage of cell abscission and may involve interaction with the Ephrin-B/RGS3 pathway, DLG5 and/or other unidentified factors present in the intercellular bridge. Knockout of the *Kif20a* gene in NPCs does not appear to compromise the completion of cytokinesis (the canonical function of KIF20A) but affects the balance between proliferation and differentiation (the cell fate determination function of KIF20A). The cartoons of illustration were modified based on Mierzwa and Gerlich[46]

cell, indicating that MR is not inherently critical for cell fate determination. In this study, we found that KIF20A was present at the midbody of the intact ICB but was not associated with the MR after cell separation. In view of our observations of the specific interaction between KIF20A and RGS3, their overlapping presence in the ICB of dividing NPCs, and their crucial importance for the maintenance of the progenitor cell fate, our data suggest that information for cell fate specification between nascent daughter cells likely resides in protein interaction networks

within the ICB. In a more broad view of stem/progenitor cell divisions, the potential of the ICB as the site of arising symmetry or asymmetry might also be useful for thinking about how asymmetry could be created during differentiative divisions in non-polarized stem/progenitor cells. Identifying additional factors of the ICB that are important for cell fate determination and understanding how these factors orchestrate the specification of daughter cell asymmetry in coordination with the cytokinesis machinery will be an important task for future studies.

## Methods

**Plasmids and antibodies.** Plasmids used in co-IP and over-expression experiments were cloned either under the control of elongation factor 1 (EF1) promoter, including EF-HA-KIF20A, EF-myc-PDZ-RGS3, EF-myc-PDZ-RGS3ΔRGS, EF-GFP-RGS, EF-HA-RBD, EF-GFP-PDZ-RGS3, and EF-GFP-PDZ-RGS3ΔRGS or under the control of CAG promoter, including CAG-RFP-KIF20A-DCX-GFP, CAG-mRFP-KIF20A, CAG-KIF20A-CDS, CAG-GFP, CAG-GFP-PDZ-RGS3-IRES-ephrin-B1. CMV-YFP-Gαi2 was a gift from John H. Kehrl. Primary antibodies included Rat anti-HA (Roche; 1:500), Mouse anti-myc clone 9E10 (Sigma M4439; 1:5000), Rabbit anti-Flag (Sigma F7425; 1:2000), Sheep anti-digoxigenin–alkaline phosphatase (Roche; 1:1000), Mouse anti-GFP (Roche; 1:200), Goat anti-KIF20A (L-13) (Santa Cruz; 1:100), Rabbit anti-Mklp1 (N-19) (Santa Cruz; 1:50), Mouse anti-α-Tubulin (Sigma T6199; 1:5000), Rabbit anti-RGS3 (1:1000)[15], Rabbit anti-Gαo2 (1:200)[23], Mouse anti-BrdU (Sigma B2531; 1:100), Mouse anti-Ki67(BD; 1:25), Mouse anti-βIII-Tubulin (Sigma T8660; 1:1000), Goat anti-Sox2 (Santa Cruz; 1:50), and Rabbit anti-activated caspase 3 (BD Biosciences; 1:200). Additional antibodies and reagents included Rabbit anti-Pax6 (Covance; 1:500), Chicken anti-Tbr2 (Abcam; 1:500), Rabbit anti-Sox5(H-90) (Santa Cruz; 1:50), Goat anti-Cux1 (C20) (Santa Cruz;1:50), Rabbit anti-NeuN (Millipore ABN78; 1:200), ConA (Molecular Probes; 1:100), and Click-iT®EdU Alexa Fluor®594 image kit (ThermoFisher Scientific). All secondary antibodies were purchased from Jackson ImmunoResearch Laboratories (Cy3, Cy5, and Cy2 AffiniPure conjugated; 1:200).

**Yeast two-hybrid screening.** A mouse E9.5/10.5 embryonic two-hybrid cDNA library[15] was screened using both full-length PDZ-RGS3 and the RGS domain alone as baits. A total of 20 million yeast cells were screened with each bait and library cDNAs from reproducibly positive colonies were recovered and sequenced. KIF20A was the candidate PDZ-RGS3 interactor that was pulled out by both screenings.

**Co-immunoprecipitation and western blot.** Plasmids were transfected into HEK293 cells using calcium phosphate method. Transfected cells were sonicated in cold lysis buffer (10 mM Tris-HCl, pH 7.5, 150 mM NaCl, 1% NP40, 1 mM EDTA, 1 mM dithiothreitol (DTT), and protease inhibitors). Either Flag or Myc antibody was used to bind the target protein at a final concentration 1 µg/ml, and then magnetic protein G beads (20398, Pierce) were used to precipitate the antibody–protein complex. For endogenous protein co-IP, 14 E15.5 brains (including cortices and ganglionic eminences) were pooled in 500 µl of lysis buffer (10 mM Tris-HCl, pH 7.5, 150 mM NaCl, 1%NP40, 1 mM EDTA, 1 mM DTT, and 1× protease inhibitor cocktail), triturated using a pipette with a blue tip, and sonicated for 30 s on ice. After incubation for 4 h on a rotation wheel at 4 °C, the lysate was centrifuged at 13,000 × g for 15 min at 4 °C. The cleared lysate was diluted with 3 volumes of dilution buffer (10 mM Tris-HCl, pH 7.5, 150 mM NaCl, 0.2% NP40, 1 mM EDTA, 1 mM DTT, 1× protease inhibitor cocktail). One half of the diluted lysate was mixed with 5 µg of anti-RGS antibody and the other half was mixed with 5 µg of control IgG, and then incubated with magnetic protein G beads. The beads containing antibody–protein complex were washed and boiled in 2× sodium dodecyl sulfate (SDS) buffer. Denatured proteins were resolved by SDS–polyacrylamide gel electrophoresis and transferred to a polyvinylidene difluoride membrane for western blot detection by horseradish peroxidase-conjugated antibody with chemiluminescent substrate (Pierce). Uncropped western blots for Fig. 2e–g can be found in Supplementary Fig. 11.

**RNA in situ hybridization and immunohistochemistry.** RNA in situ hybridization was done with digoxigenin-labeled RNA transcribed from a fragment of KIF20A cDNA 1–1065 bp followed by anti-digoxigenin alkaline phosphatase detection. For immunostaining of whole-mount cortices, E14.5 cortices were dissected out and fixed in 4% cold paraformaldehyde in phosphate-buffered solution (PBS) for 10 min. After rinsing three times with 0.5% Triton X-100 in PBS, the cortices were blocked for 2 h at room temperature in blocking solution (1× PBS, 0.5% Triton X-100, 10% donkey serum, 0.2% sodium azide), followed by incubation with primary antibody (anti-KIF20A, 1:50; anti-MKLP1, 1:100) in blocking solution at 4 °C for overnight. The cortices were rinsed three times with blocking solution followed by another three times rinse with 0.5% Triton X-100 in PBS. The cortices were then incubated with secondary antibody in blocking solution without sodium azide at 4 °C for overnight. After rinsing three times with 0.5% Triton X-

100 in PBS followed twice with PBS, the cortices were mounted to a glass slide with Fluoromount-G (SouthernBiotech) for imaging with a confocal microscope.

**shRNA design and screening.** The shRNAs were expressed under the control of a mouse U6 promoter in pNUTS vector which additionally contains an ubiquitin promoter-EGFP expression cassette. Candidate shRNAs were screened with targets cloned in psi-CHECK vector in transfected HEK293 cells, using a dual luciferase reporter assay (Promega, E1910) and verified by quantitative PCR and western blot. The sequences of 19-mer shRNAs (the loop of shRNA is underlined) used in this study were:
shKIF20A-3'UTR (5′ –GGATGAAGATGCATGGTTGTTCAAGAGACAACCATGCATCTTCATCC-3'),
shKIF20A-CDS (5'-TGGCTGAGCTGCAGAATAATTCAAGAGATTATTCTGCAGCTCAGCCA-3'),
shRGS3 (5'-AGAGAGGAAGATGTTTGAGATTCAAGAGATCTCAAACATCTTCCTCTCT-3'),
Scrambled (5'-CGGCTGAAACAAGAGTTGGTTCAAGAGACCAACTCTTGTTTCAGCCG-3').

**Establishment of Myc-KIF20A inducible cell line.** TetR was introduced into CHP100 cells using a lentiviral expression system and the infected cells were selected for stable single clones with G418 (400 µg/ml) in Dulbecco's modified Eagle's medium (DMEM; Invitrogen) with 10% fetal bovine serum. Multiple clones were tested for doxycycline inducibility using a TetO-luciferase expression system and clone #15 was selected for further use. TetO-Myc-KIF20A expression cassette was then introduced into the TetR-expressing clone #15 cells by lentiviral infection and the cells were selected for stable single clones using G418 (200 µg/ml) and puromycin (1 µg/ml) double selection in culture medium. Multiple clones were tested for induced expression of Myc-KIF20A and clone #1 was selected for subsequent experiments. Transfection of clone #1 with PDZ-RGS3 plasmids was done using TrueFect reagent (NF-0866-3, United Biosystem). Cells were induced to express Myc-KIF20A with doxycycline (0.1 µg/ml) at the time of transfection and were later fixed for immunostaining 48 h after transfection.

**Cortical primary culture, transfection, and paired cell assay.** The E14.5 mouse cortices were dissected out and dissociated by trituration in Hanks' balanced salt solution (HBSS) (Mediatech). Cells were re-suspended in culture medium DMEM with 4.5 g/l Glucose (Mediatech), glutamine (Gibco), penicillin/streptomycin (Gibco), sodium pyruvate (Gibco), 1 mM N-acetyl-L-cysteine (Sigma), B27 (Invitrogen), N2 (Invitrogen), and 10 ng/ml basic fibroblast growth factor-2 (bFGF2; Gibco), and were plated down into poly-D-lysine (P6407, Sigma)-coated 24-well glass-bottom plate (P24G-1.5-13-F, MatTek) at $2 \times 10^5$ cell per well. After 20 h of culture, cells were processed for antibody staining. The experiment was repeated twice.

**In utero electroporation and phenotype analysis.** Animal procedures were approved by the institutional animal care and use committee of Beckman Research Institute of the City of Hope and were carried out in accordance with the National Institutes of Health (NIH) guideline and the Guide for the Care and Use of Laboratory Animals. IUE was performed on E13.5 embryos and the transfected brains were dissected out 2, 3, or 4 days later. Consecutive coronal sections (14 µm) of an injected brain were collected and detected by direct visualization of GFP expressed from the shRNAs or over-expression plasmid. The center sections along the anterior-posterior axis of the fluorescent region of individual brains (six or more injected brains for each DNA plasmid combination) were used for quantification with Image-Pro Premier (Media Cybernetics).

**Live cell imaging.** Dissociate cells were prepared using the E13.5 Dcx-mRFP transgenic mouse cortices by trituration in HBSS (Mediatech). Cells were re-suspended in culture medium of DMEM with 4.5 g/l glucose and sodium pyruvate without L-glutamine and phenol red (Mediatech), supplemented with glutamine (Gibco), penicillin/streptomycin (Gibco), 1 mM N-acetyl-L-cysteine (Sigma), B27 (Invitrogen), N2 (Invitrogen), and 10 ng/ml bFGF2 (Gibco), and were plated down into poly-D-lysine (P6407, Sigma)-coated Hi-Q4 Quadruple well glass-bottom plate (MZI00040, Ibidi) at $2 \times 10^5$ cell per well. Cells were infected with Lentivirus of shKif20a or scrambled shRNA for 24 h. Live imaging of the infected cells was then taken with BioStation IM-Q (Nikon) microscopic machine at 12 min intervals for 2 days. Time-lapse images were processed by BioStation IM (Version 2.22) and Image-Pro Premier 9.2 (64-bit) Software.

**Generation of Kif20a knockout first and conditional mice.** Three Kif20a targeted ES cell clones obtained from IMPC were used for injection into blastocysts and generation of chimerical mice. One ES cell clone was able to produce chimeras.

The chimeras were crossed with wild-type C57BL/6J to generate knockout-first heterozygous (KF/+) F1 mice, which were used to expand the knockout-first strain. To produce conditional mice, F1 (KF/+) mice were crossed with ROSA26^FlpeR mice to remove the neomycin and LacZ cassette and the resultant strain became the Kif20a conditional knockout (fl/+) mice.

**Analyses of Kif20a germline and conditional knockout mice.** IUE was performed on E13.5 embryos. To ensure consistent comparison between knockout and control brains, CAG-Cremyc-2A-GFP and CAG-GFP plasmids were electroporated into embryos of the same pregnant *Kif20a*^fl/fl mice, with each plasmid being injected into embryos of one of the two uterine horns, respectively. Electroporated brains were dissected out at E16.5 (72 h post electroporation). Consecutive coronal sections (12 μm) of injected brains were collected and transfected cells were detected by direct visualization of GFP expressed from the plasmids. The center sections along the anterior–posterior axis of the fluorescent region of individual brains (five or more injected brains for each DNA plasmid) were used for quantification with Image-Pro Premier (Media Cybernetics). For acutely dissociated cortical cell culture, electroporated cortices were collected at E15.5 and dissociated cells were plated onto poly-D-lysine-coated coverslip in culture for 3 h. Cells were then fixed and stained for different markers.

For BrdU and EdU birth dating, pregnant female mice were first administered with tamoxifen in corn oil (80 mg/kg weight body) by intraperitoneal injection at E9.5 and E10.5, and then were labeled with BrdU (100 mg/kg) and EdU (100 mg/kg) sequentially at E12.5 and E15.5, respectively. The labeled pups were delivered by C-section at E19.5 for brain analysis. Cryosections of E19.5 brains (16 μm) were analyzed for immunostaining on BrdU, EdU, early-born neuron maker SOX5, and late-born neuron maker CUX1.

For FACS-based cell cycle analysis, pieces of the E15.5 cortices were gently dissociated in cold HBSS (Mediatech) with 5 mM EDTA. After centrifugation with $300 \times g$ at 4 °C for 10 min, cells were washed twice with cold HBSS buffer. Subsequently, large cell/aggregates were removed with a cell strainer (40 μm). After fixation with 70% ethanol drop by drop for overnight at 4 °C, cells were washed in cold PBS twice. For PI staining, cells were re-suspended in 0.5 ml staining buffer (PBS + 10 μg/ml PI + 0.1% Triton X-100 + 0.2 mg/ml DNAse-free RNAse A) and were incubated at 37 °C for 30 min. Cells were filtered before FACS analysis. Measurements were done using a CyAn ADP instrument (DakoCytomation) in combination with summit software.

**RNA-seq.** PolyA RNAs from the E12.5 wild-type and *Kif20a* germline or inducible knockout mutant cortical cells were isolated. RNA-seq was done by the Integrative Genomics Core at City of Hope using Illumina HiSeq2000 system following the manufacturer's protocols (Illumina). Reads were aligned to the mouse reference genome mm9 using TopHat v2.0 with default settings. The expression level of RefSeq genes were counted by matching the aligned reads to the coordinates of the RefSeq exons. Reads falling into exons that belong to multiple transcripts of the same genes were counted one time, and the reads from all exons of the same gene were combined to represent the gene level expression. Raw counts were then normalized by trimmed mean of M-value (TMM) method implemented in the Bioconductor package "edgeR". The normalized counts were then scaled by the gene length and log2 transformed to represent the expression value of each RefSeq gene model.

**Image acquisition and processing.** Fluorescent images were taken with a confocal microscope Zeiss LSM 510 Upright two photon or Olympus Inverted IX81. RNA in situ images were captured with Spot Insight GE Camera.

**Data availability.** The authors declare that all data supporting the findings of this study are available within the article and its Supplementary Information files or from the corresponding author upon reasonable request.

RNA-seq data of wild-type and *Kif20a* mutant cortical cells have been deposited in the Gene Expression Omnibus (GEO) database under accession code GSE108624.

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

## Acknowledgements

We thank Donna Isbell, Jeremy LaDou, and Cirila Arteaga for providing animal care. We thank Dr. Ryoichiro Kageyama for providing Nestin-CreERT2 mice and Dr. John H. Kehrl for providing YFP-Gαi2 expression plasmid. This work is supported by NIH grant NS096130 to Q.L.

## Author contributions

A.G., R.Q., K.M., J.L., H.Z., H.F., and N.D. conducted the experiments. X.W. performed RNA-seq analysis of wild-type and *Kif20a* knockout brain cells. M.J., J.-K.Y., and W.T. provided assistance of characterization of G protein antibodies, generation of Tet-on inducible cell lines, and *Kif20a* knockout mice. A.G., R.Q., and Q.L. analyzed data and wrote the manuscript.

## Additional information

**Competing interests:** The authors declare no competing interests.

