## [Peer Review File · Nature Communications]

Reviewers' Comments:

Reviewer #1:

Remarks to the Author:

In this study Geng and colleagues focused on the role of the mitotic kinesin KIF20A in regulating the decision of cortical neural precursors cells (NPCs) to undergo symmetric or asymmetric division, that is to generate proliferating cells or to differentiate into neurons, respectively. This is an important subject. The authors describe the interaction between KIF20A and a regulator of G protein signaling, RGS3. More specifically, they find that RGS proteins bind Ephrin receptors mediating its signaling pathway, and co-localize with KIF20A in the intracellular bridge (ICB) of early proliferating NPCs, during mouse corticogenesis (Figure 1). KIF20A is differently segregated during NPCs division and it is expressed selectively in proliferating cells but not in the sibling neurons (Figure 2 and 3). Depletion of KIF20A by RNA interference during mouse cortical development induces early differentiation of NPCs into neurons, and premature migration from the VZ to the CP (Figure 2 and 3). In addition, the authors show a coordination of KIF20A with Ephrin signaling pathway. KIF20A knockdown reverts Ephrin mediated proliferative effect and promote neuronal differentiation. Moreover, KIF20A regulates RGS3 localization in the ICB and both proteins control the cell fate specification (Figure 4). The authors have also generated KIF20A KO and conditional KO mice models. KIF20A KO pups show cortical abnormalities as thinner neuronal layers and loss of the NPCs pool expressing Pax6 or TBR2 (Figure 5), due to their early cell cycle exit and premature differentiation into neurons (Figure 6). Precocious cycle exit and neuronal differentiation of the cortical NPCs was observed also after conditional deletion of KIF20A by Cre mediated recombination, in vivo and in in vitro cultured NPCs (Figure 6, 7 and S8). The authors conclude that KIF20A controls cell fate determination and the balance between symmetric and asymmetric cell division. It plays an important role during cytokinesis through its interaction with RGS3 and G α subunit at the ICB, and it is essential to maintain the progenitor state.

The role of KIF20A in controlling NPCs fate determination is demonstrated in a robust way through experiments performed in many models. The present paper provide convincing evidences that the distinct segregation of KIF20A in siblings NPCs causes different fate choice and that the lack of KIF20A induces early cell cycle exit of the NPCs and neuronal differentiation, in vitro and in vivo. However, the work is limited to a description of the effect of the KIF20A depletion, but no mechanisms through which KIF20A regulates the cell fate choice have been investigated. Potential interesting mechanisms are instead just mentioned as speculations in the Discussion section. Moreover, the work extends vary little the basic knowledge in the field of how the choice to transition from proliferative NPCs to differentiated cells is controlled and how fate is determined. In fact, no fate determinant, nor ligands/extracellular stimuli controlling KIF20A segregation, nor transcription factors activated downstream of KIF20A, is identified in this work.

It is very important that the authors have considered the relationship between KIF20A and Ephrin signaling pathway components. However, it would be interesting and instructive to investigate also the relationship that KIF20A has with the other well known cell fate determinant signaling pathways and transcription factors, which control the proliferative state versus the neuronal differentiation of the NPCs. For instance, it would be important to understand how BMPs, WNTs, Notch-Numb, FGFs, AKT, MAPK signaling-related or cell cycle machinery (CDKs and Cyclins)-related components segregate in the sibling cells in relation to KIF20A, and define the downstream transcriptional network activated by KIF20A. Similar approaches were described in the papers, where proliferation versus neurogenesis was studied (e.g. Rash BG, J of Neuroscience 2011; Lange C, Cell Stem Cell 2009; Oishi K, PNAS 2009; Sun Y, Neuron 2005). The tools generated by the authors and used in this work, such as KIF20A germ line and conditional KOs can be used to extend the work and define the transcriptional dynamics downstream of KIF20A which can be responsible of the proliferative versus neuronal fate determination.

Minor points:

Of the first 14 references cited in the introduction, 12 are general reviews on a similar topic.

Maybe more specific works should be cited since the introduction looks vague. It is not clear in the introduction if the study has been performed to understand cortical neurogenesis or if developing cortex has been used as one of many model tissues to study symmetric versus asymmetric cell division.

Figure 2C: It is not clear what shRNA-GFP vector is represented, if the control or the KIF20A shRNA.

Figure 3: Not clear what shRNA-GFP vector is represented. Moreover, the time lapse micro-recording should be completed with the lineage dendrograms illustrating the lineage analysis

Reviewer #2:

Remarks to the Author:

The authors previously reported that regulation of G α signaling by RGS3 plays a role in a fate decision of neural progenitor cells between symmetric (self-renewal) vs asymmetric (differentiation) cell divisions. In this manuscript, prompted by their own finding of the interaction between RGS3 and a kinesin-like protein KIF20A/MKLP2, they studied the role of KIF20A in neural development. Germ-line and inducible knockout of KIF20A resulted in defects in brain development, which was attributable to mislocalisation of RGS3 and precocious cell cycle exit and neuronal differentiation, but not to the failure in cell division.

According to the data presented, it seems to be true that loss of function of kif20a results in abnormalities in brain development. Since a role of Kif20A in cytokinesis has been well established, it is crucial to clarify whether the phenotype was an indirect consequence of cytokinesis failure or it was caused by loss of a specific function of Kif20A in neuronal development such as the fate decision apart from its role in cytokinesis. However, the data for the absence of multinucleation are not very convincing. Decision of neuronal fate by localised G-protein signaling by a microtubule motor protein can potentially be interesting. However, experimental data presented are not sufficiently strong to support the authors' model.

Major points

1. Absence of cytokinesis failure in Kif20a^{-/-} cortices.

An image of a cortical slice in Figure S7C is the only data at a single cell resolution presented to claim the absence of cytokinesis failure in the cortex. In this area, actually, the majority of the cells in the Kif20a^{-/-} cortex seem to be mononucleated. However, there are large cells with multiple nuclei (or a single merged giant nucleus) at the cortical boundary (top edge), which are likely to have been generated by cytokinesis failure. Proper statistical analysis is necessary.

2. Evidence for RGS3-KIF20A interaction

2-a. Yeast 2-hybrid

First, although this interaction was identified by yeast 2-hybrid assay, no data have been presented. Actual data should be presented. Second, necessary controls for the immunoprecipitation experiments are missing.

2-b. Co-immunoprecipitation

The method to show the interaction taken by the authors uses three antibodies a) antibody for immunoprecipitation, b) primary and c) secondary antibodies for detection by western blotting. Importantly, the antibody for immunoprecipitation is bought into the samples for the SDS-PAGE and appears as major ~50 kDa and ~25 kDa band, which correspond to heavy and light chains IgG. Notably, since heavy and light chains of antibody are covalently linked by disulfide (S-S) bonds, if the dissolution of these bonds by reduction reaction is insufficient, they run as smear bands with ~75, ~100, ~125, ~150 kDa mobilities on the gel. Since they typically exist hugely excess to the antigen and co-immunoprecipitated proteins, their cross-reaction with the secondary

antibodies can be significant. Thus, for this kind of experiments, it is crucial to make sure 1) to maximise the dissolution of IgG subunits by reduction reaction, 2) to prove the identities of the detected bands by including proper controls.

Unfortunately, in Figure S1, only small parts of the western blotting were presented without any information about the mobility in SDS-PAGE (molecular weight). We can't properly examine the identities of the bands on the blotting, i.e., whether they are actually the target of IP or the co-immunoprecipitated proteins. The whole picture of the membrane should be presented.

To appropriately control the possible cross-reaction between the antibody used for immunoprecipitation and the secondary antibody for the western blotting, the control IgG must be from the same species with the one used for the immunoprecipitation (IgG control in Figure S1b, S1d and S1g should be from rabbit, the one for Figure S1e should be from rat and the one for Figure S1f should be mouse). However, the description about the sources and species of "control IgG" is totally missing.

More specifically,

(Figure S1b) It is not clear whether the "IP: IgG" lanes can be compared with "IP: Flag" lanes. The blotting of immunoprecipitates by Flag and by IgG control should be analysed side by side on the same membrane. Even from the same species, each antibodies can behave differently. Strictly, to prove the "myc-PDZ-RGS3" band in "IP: Flag"&"Blot: myc" membrane is actually myc-PDZ-RGS3, a mock immunoprecipitation from the buffer should be performed in parallel with the immunoprecipitation from the cell extract and analysed side by side.

(Figure S1c) With the images presented, we can't exclude a possibility that the band labeled "Flag-Kif20a" in "IP: myc"&"Blot: Flag" is actually myc-PDZ-RGS3 cross-reacted with the anti-Flag or secondary antibodies used for blotting. Whole the membrane should be presented. A powerful straight-forward control is to perform immunoprecipitation from the cells expressing only myc-RGS3 without flag-Kif20a in parallel with the experiment shown in the current figure.

(Figure S1d) To me, both the smear bands labeled "Flag-Kif20a" and "RGS3" look like insufficiently reduced rabbit anti-RGS3 antibody used for immunoprecipitation, which was detected by the anti-rabbit secondary antibody (both the primary antibodies for blotting are from rabbit).

In conclusion, biochemical evidence for the RGS3-Kif20a interaction is far from convincing.

Minor points

1. In Figure S1a, the coiled coil is indicated to start at amino acid 517. However, there are multiple proline residues (P528, P532 and P548) after this site. Programs such COILS (https://embnet.vital-it.ch/software/COILS_form.html) predict coiled coil start at around amino acid 550.

Reviewer #3:

Remarks to the Author:

Comments on "KIF20A/MKLP2 regulates the division modes of neural progenitor cells during cortical development", by Geng et al.

Geng and colleagues show that KIF20A influences NPC division and fate by remaining only in progenitor cells, but not postmitotic neurons. KIF20A is shown to interact with RGS3 in monolayer cells, and both are found on the midbody of dividing NPCs. Loss of KIF20A is shown to cause some NPC apoptosis, and significant NPC depletion. The data has been gathered carefully and the results suggest a function in NPC fate determination. However, exactly what that function is remains unclear. This manuscript is quite suggestive, but the conclusions are not yet sufficiently supported

for publication.

1- The evidence for a direct causal KIF20A function in neuronal differentiation is not clear. Could it simply be that missing a key mitotic gene prevents division and indirectly leads to eventual differentiation? If not, how and to what extent loss KIF20A and RGS3 would be able to directly drive neuronal differentiation?

2- Much of the early data revolves around KIF20A being only in daughter progenitors. Yet this seems hardly surprising, as purely mitotic proteins are not expected in postmitotic neurons. In fact, this elegant finding would rather argue against a non-mitotic, differentiation function for KIF20A.

3- In Fig2a and 4a it seems strange to find so few GFP cells in the CP of Scr. after 2 days of IUEP, especially when many more are present in the Fig2b Scr., but which then does not look so different to the shKif20a panels. Those key datasets appear to be in conflict and this complicates supporting a neuronal differentiation function for KIF20A. Are the authors sure that their plasmids are having no side effects?

4- On the other hand, NPC depletion on the more ventricular side when KIF20A or RGS3 are perturbed (Fig. 2a, b; 4a, 5e, etc.) seems solid and potentially very interesting. The authors themselves point out that apoptosis in early-mid neurogenesis could be a partial cause. A more detailed dissection of this depletion may help to clarify the function(s) of KIF20A.

5- Based on the very low resolution in FigS7c, it seems hard to completely discard multinucleation, especially given KIF20A's established role in cytokinesis. In addition, faulty cytokinesis could also result in chromosome missegregation and aneuploidy. Higher resolution XYZ data are needed to really discard multinucleation, and aneuploidy could at least be discussed as a possible outcome.

6- Fig6b and FigS8a need controls for proper interpretation of the data.

Response to reviewers' comments

Reviewer #1 (Remarks to the Author):

In this study Geng and colleagues focused on the role of the mitotic kinesin KIF20A in regulating the decision of cortical neural precursors cells (NPCs) to undergo symmetric or asymmetric division, that is to generate proliferating cells or to differentiate into neurons, respectively. This is an important subject. The authors describe the interaction between KIF20A and a regulator of G protein signaling, RGS3. More specifically, they find that RGS proteins bind Ephrin receptors mediating its signaling pathway, and co-localize with KIF20A in the intracellular bridge (ICB) of early proliferating NPCs, during mouse corticogenesis (Figure 1). KIF20A is differently segregated during NPCs division and it is expressed selectively in proliferating cells but not in the sibling neurons (Figure 2 and 3). Depletion of KIF20A by RNA interference during mouse cortical development induces early differentiation of NPCs into neurons, and premature migration from the VZ to the CP (Figure 2 and 3).

In addition, the authors show a coordination of KIF20A with Ephrin signaling pathway. KIF20A knockdown reverts Ephrin mediated proliferative effect and promote neuronal differentiation. Moreover, KIF20A regulates RGS3 localization in the ICB and both proteins control the cell fate specification (Figure 4). The authors have also generated KIF20A KO and conditional KO mice models. KIF20A KO pups show cortical abnormalities as thinner neuronal layers and loss of the NPCs pool expressing Pax6 or TBR2 (Figure 5), due to their early cell cycle exit and premature differentiation into neurons (Figure 6). Precocious cycle exit and neuronal differentiation of the cortical NPCs was observed also after conditional deletion of KIF20A by Cre mediated recombination, in vivo and in vitro cultured NPCs (Figure 6, 7 and S8). The authors conclude that KIF20A controls cell fate determination and the balance between symmetric and asymmetric cell division. It plays an important role during cytokinesis through its interaction with RGS3 and G α subunit at the ICB, and it is essential to maintain the progenitor state.

The role of KIF20A in controlling NPCs fate determination is demonstrated in a robust way through experiments performed in many models. The present paper provide convincing evidences that the distinct segregation of KIF20A in siblings NPCs causes different fate choice and that the lack of KIF20A induces early cell cycle exit of the NPCs and neuronal differentiation, in vitro and in vivo. However, the work is limited to a description of the effect of the KIF20A depletion, but no mechanisms through which KIF20A regulates the cell fate choice have been investigated. Potential interesting mechanisms are instead just mentioned as speculations in the Discussion section. Moreover, the work extends very little the basic knowledge in the field of how the choice to transition from proliferative NPCs to differentiated cells is controlled and how fate is determined. In fact, no fate determinant, nor ligands/extracellular stimuli controlling KIF20A segregation, nor transcription factors activated downstream of KIF20A, is identified in this work.

It is very important that the authors have considered the relationship between KIF20A and Ephrin signaling pathway components. However, it would be interesting and instructive to investigate also the relationship that KIF20A has with the other well known cell fate determinant signaling pathways and transcription factors, which control the proliferative state versus the neuronal differentiation of the NPCs. For instance, it would be important to understand how BMPs, WNTs, Notch-Numb, FGFs, AKT, MAPK signaling-related or cell cycle machinery (CDKs and Cyclins)-related components segregate in the sibling cells in relation to KIF20A, and define the downstream transcriptional network activated by KIF20A. Similar approaches were described in the papers, where proliferation versus neurogenesis

was studied (e.g. Rash BG, J of Neuroscience 2011; Lange C, Cell Stem Cell 2009; Oishi K, PNAS 2009; Sun Y, Neuron 2005). The tools generated by the authors and used in this work, such as KIF20A germ line and conditional KOs can be used to extend the work and define the transcriptional dynamics downstream of KIF20A which can be responsible of the proliferative versus neuronal fate determination.

Thank you for your positive comments.

We agree with the assessment that the mechanisms by which KIF20A acts to regulate cell division modes are not well defined. By virtue of the presence of all involved fate regulators in this molecular axis, including RGS3, KIF20A, and Galpha subunit, in the ICB during cytokinesis, it is conceivable that cell fate specification in the two nascent daughter cells occurs when they become independent by cleavage, during which time microtubule-associated motors, such as KIF20A, may help create symmetric or asymmetric distribution of cell fate determinants in the two daughter cells, which in turn will lead to augmentation of the commitment to either continued proliferation or differentiation. Dynamically expressed transcriptional networks would most likely be the regulatory cues for the commitment of the final fate.

As suggested, we performed experiments to examine genes activated or suppressed in the mouse cortex as a result of LOF of KIF20A. Because tamoxifen-induced Cre-loxP system may show variable patterns of Cre expression in the target tissue cells (e.g. mosaic expression of Cre), which may affect gene expression studies, we have performed transcriptome analyses using cortical tissues derived from the germline knockout mice. RNAs of three batches of E12.5 wild-type (WT) or mutant (KO) cortices were sequenced by deep sequencing. The raw data were deposited with GEO. As summarized in a new supplemental Figure S9, the sequencing data identified 80 down-regulated and 193 up-regulated genes (Fold change FC \geq 1.5; P value \leq 0.05) in the mutant cortices. While these differentially expressed genes were in general having functions consistent with the change of state of NPCs from proliferation to differentiation (gene ontology analysis shown in Figure S9c), the sequencing data could not confidently reveal any significant signaling pathways or family of molecules that are specifically associated downstream of KIF20A function. We suspect this might be due to a transient nature of KIF20A's action/function in cell fate regulation, which should occur at the late phase of cytokinesis. One possible scenario is that KIF20A helps facilitate distribution of general cell fate regulator-containing cargos (rather than cell fate regulators of specific families) into the two nascent daughter cells, which may occur symmetrically or asymmetrically. Once the cell abscission is completed, the resulting symmetrical or asymmetrically inherited cargo contents may in turn trigger signals for the final fate commitment to either continued proliferation or differentiation (perhaps via establishing a proliferation state or differentiation state of transcriptome). We have discussed such a potential mechanism in the discussion section.

Minor points:

Of the first 14 references cited in the introduction, 12 are general reviews on a similar topic. Maybe more specific works should be cited since the introduction looks vague. It is not clear in the introduction if the study has been performed to understand cortical neurogenesis or if developing cortex has been used as one of many model tissues to study symmetric versus asymmetric cell division.

We have revised the introduction section to replace most of the general reviews with some specific works and discussed on cell fate regulators identified from the invertebrate studies in relation to cell fate determination in the mouse cerebral cortical development.

Figure 2C: It is not clear what shRNA-GFP vector is represented, if the control or the KIF20A shRNA.

We have added this information to the Figure legend (page 36).

Figure 3: Not clear what shRNA-GFP vector is represented. Moreover, the time lapse micro-recording should be completed with the lineage dendrograms illustrating the lineage analysis

We have added the shRNA-GFP vector information to the Figure legend (page 37).

Lineage trees of some representative divisions of the live cell imaging data were included in new Figure 3b-g.

Reviewer #2 (Remarks to the Author):

The authors previously reported that regulation of G α signaling by RGS3 plays a role in a fate decision of neural progenitor cells between symmetric (self-renewal) vs asymmetric (differentiation) cell divisions. In this manuscript, prompted by their own finding of the interaction between RGS3 and a kinesin-like protein KIF20A/MKLP2, they studied the role of KIF20A in neural development. Germ-line and inducible knockout of KIF20A resulted in defects in brain development, which was attributable to mislocalisation of RGS3 and precocious cell cycle exit and neuronal differentiation, but not to the failure in cell division.

According to the data presented, it seems to be true that loss of function of *kif20a* results in abnormalities in brain development. Since a role of *Kif20A* in cytokinesis has been well established, it is crucial to clarify whether the phenotype was an indirect consequence of cytokinesis failure or it was caused by loss of a specific function of *Kif20A* in neuronal development such as the fate decision apart from its role in cytokinesis. However, the data for the absence of multinucleation are not very convincing. Decision of neuronal fate by localised G-protein signaling by a microtubule motor protein can potentially be interesting. However, experimental data presented are not sufficiently strong to support the authors' model.

Major points

1. Absence of cytokinesis failure in *Kif20a*^{-/-} cortices.

An image of a cortical slice in Figure S7C is the only data at a single cell resolution presented to claim the absence of cytokinesis failure in the cortex. In this area, actually, the majority of the cells in the *Kif20a*^{-/-} cortex seem to be mononucleated. However, there are large cells with multiple nuclei (or a single merged giant nucleus) at the cortical boundary (top edge), which are likely to have been generated by cytokinesis failure. Proper statistical analysis is necessary.

We agree that more data are needed to support our conclusion that LOF of KIF20A did not compromise cytokinesis per se. Within the proliferating zone (the ventricular and subventricular zones) of E12.5 or E15.5 cortical sections of the *Kif20a* knockout mice, we did not observe any obvious bi- or multi-nucleated cells. The large cells with apparent "multiple nuclei (or a single merged giant nucleus) at the cortical boundary (top edge)", as you pointed out, could be a result of artificial appearance due to folding of the meninges of the section when it landed on the slide, as this sometimes occurs with the edges of a section. To further address the issue of cytokinesis defect in an unbiased and quantitative manner, we used an alternative approach. In

this experiment, we collected cortical cells from littermates of *Kif20a* conditional knockout embryos and performed FACS analyses to directly measure cell cycle status of mutant and wild-type cells, which could also reflect on aneuploidy status of the cells. The new data (Figure 6d) showed that there was no significant difference in DNA content (both 2N and 4N) between the mutant (*Nestin-CreERT2; Kif20a^{fl/fl}*) and wild-type (*Kif20a^{fl/fl}*) cortical cells, suggesting that no obvious cell division defect was associated with *Kif20a* knockout. In addition, in our time lapse imaging experiments on sh*Kif20a*-expressing NPCs, we did not observe any obvious defect in cytokinesis during active cell division. Therefore, these data together demonstrated that LOF of KIF20A in the brain or in primary NPCs does not cause a defect in cytokinesis. These data did not rule out a function of KIF20A in cytokinesis, but rather suggested the following two aspects: (1) the function of KIF20A in cytokinesis of NPCs was apparently compensated in the mutant brains; (2) KIF20A is a cell fate regulator independent of its role in cytokinesis. We have discussed about these issues in the Discussion section.

2. Evidence for RGS3-KIF20A interaction

2-a. Yeast 2-hybrid

First, although this interaction was identified by yeast 2-hybrid assay, no data have been presented. Actual data should be presented. Second, necessary controls for the immunoprecipitation experiments are missing.

We presented one of the yeast two-hybrid validation results confirming some of the positive clones (Preys) identified by PDZ-RGS3 full-length protein (Bait) in a new Figure S1. Figure S1 showed that #62 (KIF20A) and #64 of this group of 50 clones obtained from the primary screening were positive for interaction with RGS3 in a secondary yeast-mating experiment (Left panel), using pLex-lamin (Bait) as a negative control for all clones (right panel). In addition, pLex-Da + pVP16-MyoD mating (blue arrows) was the positive control for protein-protein interaction which give rise to β Gal expression. pLex-RGS3 + pVP16 mating (white arrows) was another negative control.

2-b. Co-immunoprecipitation

The method to show the interaction taken by the authors uses three antibodies a) antibody for immunoprecipitation, b) primary and c) secondary antibodies for detection by western blotting. Importantly, the antibody for immunoprecipitation is bought into the samples for the SDS-PAGE and appears as major ~50 kDa and ~25 kDa band, which correspond to heavy and light chains IgG. Notably, since heavy and light chains of antibody are covalently linked by disulfide (S-S) bonds, if the dissolution of these bonds by reduction reaction is insufficient, they run as smear bands with ~75, ~100, ~125, ~150 kDa mobilities on the gel. Since they typically exist hugely excess to the antigen and co-immunoprecipitated proteins, their cross-reaction with the secondary antibodies can be significant. Thus, for this kind of experiments, it is crucial to make sure 1) to maximise the dissolution of IgG subunits by reduction reaction, 2) to prove the identities of the detected bands by including proper controls.

Unfortunately, in Figure S1, only small parts of the western blotting were presented without any information about the mobility in SDS-PAGE (molecular weight). We can't properly examine the identities of the bands on the blotting, i.e., whether they are actually the target of IP or the co-immunoprecipitated proteins. The whole picture of the membrane should be presented.

For co-IP of full-length proteins, we presented the whole picture of Western blot membranes in new Figure S2b and S2c to replace the original regional images of the blot. Multiple additional

bands of faster motility in these blots were most likely degradation products of the full length protein (as seen in the input lanes, comparing the wild-type vs. RGS domain deletion mutant lanes), some of which could be precipitated by IP. The side-by-side myc-PDZ-RGS3 and myc-PDZ-RGS3 Δ RGS lanes in S2b and S2c were mutually controls for each other in the reciprocal co-IP, because deleting the RGS domain led to failed interaction (co-IP lanes) without a difference in the expression level between the wild-type and mutant protein (input lanes). Using a deletion mutant was intended for a stronger control than an IgG control alone, as both full-length and deletion mutant were precipitated by the same antibody, but only the full-length could be properly immunoprecipitated.

Both the original Figure S1 and the current new Figure S2 supported the same conclusion that full-length RGS3 and KIF20A could be co-immunoprecipitated in a reciprocal manner.

To appropriately control the possible cross-reaction between the antibody used for immunoprecipitation and the secondary antibody for the western blotting, the control IgG must be from the same species with the one used for the immunoprecipitation (IgG control in Figure S1b, S1d and S1g should be from rabbit, the one for Figure S1e should be from rat and the one for Figure S1f should be mouse). However, the description about the sources and species of “control IgG” is totally missing.

As you correctly pointed out, in all of our co-IP experiments, the paired IP antibody (for IP of epitope tag or endogenous protein) and the blotting antibody (for Western blot) were matched from different species. Also, the HRP-conjugated secondary antibodies were manufactured to have minimal species cross reactivity to help minimize (if not completely avoid) detection of IgG heavy or light chains of the IP antibody. For instance, in the current Figure S2b, we used rabbit-anti-Flag to pull down Flag-Kif20a, and used mouse anti-myc to probe the blot. Secondary HRP-conjugated anti-mouse antibody had minimal cross reactivity with rabbit antibody.

The control IgG used in each IP was matched to the species of primary IP antibody of the experiment. For instance, Figure S2b left panel, control IgG was from rabbit to match rabbit anti-Flag antibody in the IP. We have added a description of this point to the figure legends of Figure S2b/c and S2d.

More specifically,

(Figure S1b) It is not clear whether the “IP: IgG” lanes can be compared with “IP: Flag” lanes. The blotting of immunoprecipitates by Flag and by IgG control should be analysed side by side on the same membrane. Even from the same species, each antibodies can behave differently. Strictly, to prove the “myc-PDZ-RGS3” band in “IP:Flag”&”Blot: myc” membrane is actually myc-PDZ-RGS3, a mock immunoprecipitation from the buffer should be performed in parallel with the immunoprecipitation from the cell extract and analysed side by side.

(Figure S1c) With the images presented, we can't exclude a possibility that the band labeled "Flag-Kif20a" in “IP: myc”&”Blot: Flag” is actually myc-PDZ-RGS3 cross-reacted with the anti-Flag or secondary antibodies used for blotting. Whole the membrane should be presented. A powerful straight-forward control is to perform immunoprecipitation from the cells expressing only myc-RGS3 without flag-Kif20a in parallel with the experiment shown in the current figure.

As you pointed out, the IPs by Flag and by IgG control were analyzed side by side on the same membrane (please see the whole membrane blot in new Figure S2b). In the original Figure S1, we arranged it separately simply to illustrate different lanes more clearly.

The side-by-side myc-PDZ-RGS3 and myc-PDZ-RGS3 Δ RGS lanes in Figure S2b and S2c were the ideal pairs with a natural internal control. For instance with Figure S2b, in the right panel, we could see Flag-IP brings down similar amount of Flag-Kif20a in both cell extracts, however, in the left panel, only myc-PDZ-RGS3 full-length could be effectively co-precipitated, even though myc-PDZ-RGS3 and myc-PDZ-RGS3 Δ RGS were equally expressed in the cells (judged by looking at the input lanes in the left panel). This provided solid evidence that full-length PDZ-RGS3 and Kif20a could be co-immunoprecipitated and the RGS domain was required for the interaction.

(Figure S1d) To me, both the smear bands labeled “Flag-Kif20a” and “RGS3” look like insufficiently reduced rabbit anti-RGS3 antibody used for immunoprecipitation, which was detected by the anti-rabbit secondary antibody (both the primary antibodies for blotting are from rabbit).

In conclusion, biochemical evidence for the RGS3-Kif20a interaction is far from convincing.

Endogenous proteins may be regulated with cell type-specific post-translational modifications that may not occur in the transfected cells, which can result in different moieties of the same protein with varying motilities in electrophoresis. We suspect that RGS3 and Kif20a, with both proteins over 120KDa, might be so modified, such as phosphorylated, glycosylated, etc., which may lead to the appearance of smear bands. The left panel was done using rabbit anti-RGS3 for IP and goat anti-Kif20a for blot. The first two lanes were anti-RGS3 IP and control IgG IP lanes, whereas the third lane was using Hek cell extract transfected with Flag-Kif20a as a size control. We have added descriptions to the figure legend to clarify this.

Minor points

1. In Figure S1a, the coiled coil is indicated to start at amino acid 517. However, there are multiple proline residues (P528, P532 and P548) after this site. Programs such COILS (https://embnet.vital-it.ch/software/COILS_form.html) predict coiled coil start at around amino acid 550.

This is a good point. Thank you. We have revised it to 550 in new Figure S2a.

Reviewer #3 (Remarks to the Author):

Comments on “KIF20A/MKLP2 regulates the division modes of neural progenitor cells during cortical development”, by Geng et al.

Geng and colleagues show that KIF20A influences NPC division and fate by remaining only in progenitor cells, but not postmitotic neurons. KIF20A is shown to interact with RGS3 in monolayer cells, and both are found on the midbody of dividing NPCs. Loss of KIF20A is shown to cause some NPC apoptosis, and significant NPC depletion. The data has been gathered carefully and the results suggest a function in NPC fate determination. However, exactly what that function is remains unclear. This manuscript is quite suggestive, but the conclusions are not yet sufficiently supported for publication.

1- The evidence for a direct causal KIF20A function in neuronal differentiation is not clear. Could it

simply be that missing a key mitotic gene prevents division and indirectly leads to eventual differentiation? If not, how and to what extent loss KIF20A and RGS3 would be able to directly drive neuronal differentiation?

This is a good question. KIF20A has been extensively studied in the context of cytokinesis. Although it was initially proposed to function as a Golgi stacking protein, later studies supported the notion that KIF20A is critical for correct localization of multiple key regulators, such as Aurora B, into the midbody during cytokinesis. Disruption of KIF20A function in cultured cells (the most widely used were Hela cells) was reported to lead to defect of cytokinesis (creating binucleated cells). So, it is a natural question whether what we saw in the mutant brains was an indirect result of defect in division. Our collective data demonstrated that this should not be the case. First, we did not observe any obvious buildup of bi- or multi-nucleated cells in the germline or conditional knockout mutant cortices within the proliferating zones (Figure 6c). This was also supported by new data examining the aneuploidy status of knockout cells using FACS-based cell cycle analysis (Figure 6d). Second, IUE-based knockdown (Figure 2a) or somatic knockout (Figure 7c) of KIF20A did not cause defect of divisions. In fact, the affected cells could be observed to complete the divisions but the progeny became neurons which conspicuously migrated out of the VZ. Third, in our live cell imaging of active NPC divisions under knockdown of KIF20A, we did not observe any obvious stall or defect in completion of the divisions recorded. These data collectively suggested that KIF20A in primary NPCs has a role in fate regulation independent of its role in cytokinesis.

We can only speculate that KIF20A's role in cell fate determination occurs during the late stage of cytokinesis and it most likely involves its motor function which leads to symmetric or asymmetric segregation of cell fate determinants-containing cargos between the two nascent cells. A clear picture of the precise mechanism will require identification of additional players that reside in the ICB and work with KIF20A. In this regard, we recently searched for additional factors that KIF20A may interact with and identified a KIF20A-binding ICB protein that appeared to be also involved in cell fate determination. We hope this new interaction, as well as other ICB proteins to be discovered in the future, will lead to a better understanding of the machinery that controls cell fate determination during cytokinesis.

2- Much of the early data revolves around KIF20A being only in daughter progenitors. Yet this seems hardly surprising, as purely mitotic proteins are not expected in postmitotic neurons. In fact, this elegant finding would rather argue against a non-mitotic, differentiation function for KIF20A.

This is a good point. At the time scale of our current experiment (looking at paired cells some hours after divisions have been completed), we could not tell how KIF20A (or perhaps the RGS-KIF20A complex) was segregated into the two nascent daughter cells at the time of division/cleavage. One conceivable scenario is that symmetric or asymmetric inheritance of the RGS-KIF20A complex, and its attached cargos as a result, between two nascent daughter cells at the time of ICB cleavage, could be the starting point of fate-choice signals, which will be amplified as time goes by after division. So, while we don't know precisely how KIF20A achieve its role in cell fate regulation, the observation of KIF20A associated with mitotic cells does not argue against a differentiation function for KIF20A, but we will need to consider in the context that KIF20A regulates the balance between proliferation and differentiation, where small changes such as a possible differential inheritance of KIF20A might tip the balance from a proliferation fate to a differentiation fate in the daughter cells. We have revised our discussion of the potential mechanisms of Kif20a's function to reflect these thoughts.

3- In Fig2a and 4a it seems strange to find so few GFP cells in the CP of Scr. after 2 days of IUEP, especially when many more are present in the Fig2b Scr., but which then does not look so different to the shKif20a panels. Those key datasets appear to be in conflict and this complicates supporting a neuronal differentiation function for KIF20A. Are the authors sure that their plasmids are having no side effects?

In our experience with IUE, we noticed there were variations of IUE patterns in relation to the stage of brains electroporated. While we try to maintain all electroporations at the time of E13.5, it happens that some litters of embryos show appearance of older ages/sizes resembling E14 or E14.5 stages even though the plug date was correct (this could be due to the variation of the exact mating time of the breeding pairs). In most cases, we electroporate two different plasmids into the neighboring horns of embryos of the same litter to minimize batch variation. In addition, the quantification is focused to relative distribution of each zone cells in the entire population of GFP-labeled cells rather than looking at discrete numbers of cells within each zone.

In this regard, Figure 4a and 2a/2b are not directly comparable, as Figure 4a was the injection of shRNA + RGS-ephrinB expression plasmid whereas Figure 2a/2b was injection of shRNA alone. Thus, Figure 4a had only half of the concentration of shRNA (Scr.) electroporated, and the green cell patterns would also have reflected mixed RGS-ephrinB expression plasmids.

4- On the other hand, NPC depletion on the more ventricular side when KIF20A or RGS3 are perturbed (Fig. 2a, b; 4a, 5e, etc.) seems solid and potentially very interesting. The authors themselves point out that apoptosis in early-mid neurogenesis could be a partial cause. A more detailed dissection of this depletion may help to clarify the function(s) of KIF20A.

By comparing pairs of electroporation between control (either Scr or control GFP protein) and shRNA for KIF20A or over-expression of RGS-ephrinB, we consistently observed significant differences in the distribution of green cells within the three zones of the cortices. We interpret these data as an indication of induced (in the case of knockdown) or disrupted (in the case of over-expression) neuronal differentiation and subsequent outward migration without defect in cell divisions. We only observed subtle increase of apoptosis at early stage of cortical neurogenesis, which appeared not able to account for the significant loss of NPCs. This conclusion was tested further with alternative approaches, for example, Figure 7c showed that IUE of Cre expression to somatically knockout the Kif20a gene could induce the same phenotype as seen in the case of RNAi. In addition, knockout analyses shown in Figure 7a and 8c further indicated that LOF of KIF20A led to cell cycle exit and early depletion of NPCs. We have revised the discussion section to better incorporate all of these different data to support our conclusion that LOF of KIF20A leads to precocious neuronal differentiation which depletes NPCs in the cortex.

5- Based on the very low resolution in Fig57c, it seems hard to completely discard multinucleation, especially given KIF20A's established role in cytokinesis. In addition, faulty cytokinesis could also result in chromosome missegregation and aneuploidy. Higher resolution XYZ data are needed to really discard multinucleation, and aneuploidy could at least be discussed as a possible outcome.

As the VZ cells display interkinetic nuclear migration during different phases of the cell cycle, cells around the apical surface may show variability of nuclear shapes, which may complicate the detection of bi-nucleated cells even with XYZ data, we therefore performed an additional experiment to better quantitatively address this issue in an unbiased manner. In this experiment,

we collected cortical cells from littermates of *Kif20a* conditional knockout embryos and performed FACS analyses to directly measure cell cycle status of mutant and wild-type cells, which could also reflect on aneuploidy status of the cells. Our new data (Figure 6d) showed that there was no significant difference in DNA content (both 2N and 4N) between the mutant (*Nestin-CreERT2; Kif20a^{fl/fl}*) and wild-type (*Kif20a^{fl/fl}*) cortical cells, further indicating that no obvious cell division defect was associated with *Kif20a* knockout. In addition, in our time lapse imaging experiments on sh*Kif20a*-expressing NPCs, we did not observe any obvious defect in cytokinesis during active cell division. Furthermore, knockdown (Figure 2a) or somatic knockout of KIF20A (Figure 7c) in the cortex caused the affected NPCs to differentiate into neurons which subsequently migrated out of the VZ (reflecting normal completion of NPC divisions). Therefore, all these data collectively demonstrated that LOF of KIF20A in the brain or in primary NPCs does not cause an obvious defect in cytokinesis.

6- Fig6b and FigS8a need controls for proper interpretation of the data.

Fig7b (original Fig6b) and FigS8a were intended to address the question whether re-distributions of the affected NPCs into non-proliferating zones (the IZ and CP) due to knockout of KIF20A were consistent with induced differentiation (differentiation + outward migration) or were the results of relocation of NPCs (non-differentiated but displaced as shown in some other mutant mice such as LGN/GPSM2 knockout). In the wild-type cortical sections, cells in the IZ and CP were almost invariably neurons at this embryonic stage. So, in these panels, we were looking at the mutant (knockout) brains per se. We have added further description the text to clarify this point (pages 14 and 15).

Reviewers' Comments:

Reviewer #1:

Remarks to the Author:

The authors have responded satisfactorily to this reviewer's concerns.

Reviewer #2:

Remarks to the Author:

1. Absence of cytokinesis failure in Kif20a^{-/-} cortices.

They performed FACS analyses of Nestin-CreERT2;Kif20A^{fl/fl} mutants. In principle, this is the right approach. However, this has not been performed in a proper manner that can be compared with the neuronal phenotypes. For the FACS analysis, the knock-out of KIF20A was triggered by tamoxifen at E9.5 and E10.5 and the cortical cells were collected at E15.5. However, it is unclear whether the KIF20A knock-out by this protocol can cause neuronal differentiation phenotypes at E15.5. In Figure 8, the same mice were treated with the inducer at E9.5 and E10.5 but examined for the differentiation phenotypes at E19.5. We can't exclude the possibility that the differentiation phenotype might not be so obvious at E15.5 or that the multinucleation might be detected at E19.5. FACS analysis after 10 days of KO-induction should be performed (or, if there were specific reasons to stick to E15.5, shRNA-based approach as in Figure 2 in combination with dual color analysis, i.e., GFP for transfected cells and DNA staining for DNA content should be taken).

2-a. Yeast 2-hybrid

The pattern of the colony of #R64:Kif20a looks unusual (tiny blue subcolonies appeared at the periphery of the main streak) and different from the positive controls, which show a uniform blue signal. Can the clone grow in the absence of histidine (and in the presence of 3-aminotriazole when the background growth is significant)?

2-b. Co-immunoprecipitation

I appreciate the authors' argument that side-by-side comparison between myc-PDZ-RGS3 and myc-PDZ-RGS3 Δ RGS works as mutual controls (Figure S2 b and c). Considering the results in Fig. S2e-g, too, it is likely that over-expressed Kif20a and RGS3 can show some interaction. However, the evidence for the interaction under more physiological conditions is very weak. No evidence for the interaction between endogenous molecules has been shown. Co-IP of exogenously expressed Flag-KIF20a with the endogenous RGS3 was attempted in Fig S2d. However, with the current data, we can't exclude the possibility that the smear band on the IP:RGS3 lanes is derived from the anti-RGS3 antibody used for IP. Using antibodies from the same species is not sufficient. For example, there is no proof that the IgG control was actually loaded at the same amount as that of the anti-RGS3 antibody. As I requested previously, it needs to be confirmed that this smear band disappears when the same IP experiment is performed with the buffer instead of cell lysate as an input.

By the way, I don't still understand why the molecular weight markers are not shown.

Reviewer #3:

Remarks to the Author:

Comments on the revised version of "KIF20A/MKLP2 regulates the division modes of neural progenitor cells during cortical development", by Geng et al.

Geng and colleagues have made an effort to answer the reviewer's questions and have added some important data to clarify some of the points.

1) (former point 5) In particular, the addition of cell-sorting data is important to support their

claim that cytokinesis failure does not play a critical role in the phenotypes they describe. I now regard this point as satisfied. It is not impossible that cytokinesis plays some role, but it seems unlikely to be a central role.

2) (Former points 1 & 2) Also, the gene expression analysis to look for transcriptional changes in KIF20A mutants is a step in the right direction, even though no clear mechanistic clue resulted from this analysis. The authors may nevertheless wish to repeat their analysis, but with the induced mutant tissue. The detection of transcription differences may be facilitated by an acute genotype, whereas compensatory mechanisms could cloud them in germline mutants. Nevertheless, a central criticism of this manuscript remains: There is no solid indication of what role, or even what kind of role, KIF20A may play to modify the fate of neural progenitors. For a journal with a general interest readership, it seems that at least a clear indication of the potential mechanism is necessary to solidify the presumption that KIF20A is actively involved in progenitor fate determination.

3) I understand that there is of course variation in IUE experiments. Nevertheless, it is the responsibility of the authors to ensure that experiments are made precisely at the reported stages, or the results will carry significant doubt. Also, my point remains that 48h are definitely enough for a substantial amount of electroporated wt progenitors to differentiate and migrate to the CP. This means that scr experiments should look like Fig.2b Scr., rather than Fig.2a Scr. or Fig.4a GFP. This being the case, the difference between Scr. and shKIF20A would not appear to be so significant. In addition, these variations in the Scr. data argue that your GFP plasmids and IUE procedure may be themselves responsible and need to be thoroughly controlled and perhaps replaced. Otherwise, significant portions of the main manuscript conclusions are not clearly interpretable.

In sum, while improvements have definitely been made, these still seem insufficient to justify publication.

Response to reviewers' comments

Reviewer #2 (Remarks to the Author):

1. Absence of cytokinesis failure in Kif20a^{-/-} cortices.

They performed FACS analyses of Nestin-CreERT2;Kif20A^{fl/fl} mutants. In principle, this is the right approach. However, this has not been performed in a proper manner that can be compared with the neuronal phenotypes. For the FACS analysis, the knock-out of KIF20A was triggered by tamoxifen at E9.5 and E10.5 and the cortical cells were collected at E15.5. However, it is unclear whether the KIF20A knock-out by this protocol can cause neuronal differentiation phenotypes at E15.5. In Figure 8, the same mice were treated with the inducer at E9.5 and E10.5 but examined for the differentiation phenotypes at E19.5. We can't exclude the possibility that the differentiation phenotype might not be so obvious at E15.5 or that the multinucleation might be detected at E19.5. FACS analysis after 10 days of KO-induction should be performed (or, if there were specific reasons to stick to E15.5, shRNA-based approach as in Figure 2 in combination with dual color analysis, i.e., GFP for transfected cells and DNA staining for DNA content should be taken).

We performed FACS-based cell cycle analyses at E15.5 mainly because that this stage of the cortices from each littermate embryo could provide sufficient numbers of total cells for FACS procedure and that it contains a detectable amount of dividing cells (the smaller peak of mitotic cells, G2/M cells). At E19.5, when the neurogenesis is pretty much tapered down, almost no G2/M cells from the cortex were available for detection. One example of the FACS profile from an E19.5 cko mutant brain was shown below.

To address your concern, we analyzed phenotypes of the inducible Kif20a knockout brains at E15.5. The obtained data were summarized in a new Supplemental Figure S8 (text description on page 13). The E15.5 conditional knockouts of Kif20a displayed a noticeable phenotype of early cell cycle exit and loss of cortical neural progenitor cells, which was similar to what we have observed in the germ line mutant brains at E15.5 and was consistent with the overall inducible knockout phenotype at E19.5.

2-a. Yeast 2-hybrid

The pattern of the colony of #R64:Kif20a looks unusual (tiny blue subcolonies appeared at the periphery of the main streak) and different from the positive controls, which show a uniform blue signal. Can the clone grow in the absence of histidine (and in the presence of 3-aminotriazole when the background growth is significant)?

The grainy appearance of colonies within the mating patch reflected the un-uniform time when the mated colonies were generated. The positive control patch similarly displayed the uneven blue colonies, however, the positive control pair of interaction was such a strong pair of interactor that even later mated colonies showed higher levels of blue color at the time when

pictures were taken (in the case of positive controls, the earlier mated colonies were probably saturated with blue colors so the contrast of different levels of blueness appeared to be blurred, although the grainy unevenness was still visible). Importantly, in our experience this assay produced reproducible patterns in multiple testing for candidate clones of RGS-interactors, such as #64 and #62.

2-b. Co-immunoprecipitation

I appreciate the authors' argument that side-by-side comparison between myc-PDZ-RGS3 and myc-PDZ-RGS3 Δ RGS works as mutual controls (Figure S2 b and c). Considering the results in Fig. S2e-g, too, it is likely that over-expressed Kif20a and RGS3 can show some interaction. However, the evidence for the interaction under more physiological conditions is very weak. No evidence for the interaction between endogenous molecules has been shown. Co-IP of exogenously expressed Flag-KIF20a with the endogenous RGS3 was attempted in Fig S2d. However, with the current data, we can't exclude the possibility that the smear band on the IP:RGS3 lanes is derived from the anti-RGS3 antibody used for IP. Using antibodies from the same species is not sufficient. For example, there is no proof that the IgG control was actually loaded at the same amount as that of the anti-RGS3 antibody. As I requested previously, it needs to be confirmed that this smear band disappears when the same IP experiment is performed with the buffer instead of cell lysate as an input.

By the way, I don't still understand why the molecular weight markers are not shown.

We performed co-IP of endogenous RGS3 and KIF20A with embryonic brain lysates including the conditions you suggested. The results shown in updated Figure S2d indicated that RGS3 antibody could specifically co-precipitate a band of protein reactive to KIF20A antibody. IgG heavy chains were retained in the gel to mark its presence in all samples. In addition, molecular weights of pre-stained markers were shown.

Reviewer #3 (Remarks to the Author):

Comments on the revised version of "KIF20A/MKLP2 regulates the division modes of neural progenitor cells during cortical development", by Geng et al.

Geng and colleagues have made an effort to answer the reviewer's questions and have added some important data to clarify some of the points.

1) (former point 5) In particular, the addition of cell-sorting data is important to support their claim that cytokinesis failure does not play a critical role in the phenotypes they describe. I now regard this point as satisfied. It is not impossible that cytokinesis plays some role, but it seems unlikely to be a central role.

Our data did collectively suggest that cytokinesis per se is not a major determinant to cell fate determination (or, defect in cytokinesis does not by default lead to a cell fate change), some new function of KIF20A during the late phase of cell division appeared to be contributing to cell fate determination.

2) (Former points 1 & 2) Also, the gene expression analysis to look for transcriptional changes in KIF20A mutants is a step in the right direction, even though no clear mechanistic clue resulted from

this analysis. The authors may nevertheless wish to repeat their analysis, but with the induced mutant tissue. The detection of transcription differences may be facilitated by an acute genotype, whereas compensatory mechanisms could cloud them in germline mutants.

Nevertheless, a central criticism of this manuscript remains: There is no solid indication of what role, or even what kind of role, KIF20A may play to modify the fate of neural progenitors. For a journal with a general interest readership, it seems that at least a clear indication of the potential mechanism is necessary to solidify the presumption that KIF20A is actively involved in progenitor fate determination.

As suggested, we repeated the RNA-seq analyses using cortical cells of the inducible knockout mice. The raw data from a litter of two ckoWT and three ckoMT samples were deposited in GEO. The summarized data shown in a revised Figure S10 (text revision on pages 17 and 18) identified a smaller number of up- or down-regulated genes than from the cortical cells of the germ line knockout mice, which might be consistent with the latter likely containing more secondarily affected genes. Unfortunately, inducible knockout did not seem to point to any convincing directions of specific gene families or functions that could be confidently followed up for further study either. This might be due to the heterogeneity of Cre-mediated deletion (varied levels of Cre-expression in different NPCs after tamoxifen induction which may lead to mosaic patterns of gene deletions or heterogeneity of cells) and/or the possibility that a direct effect on gene expression patterns may not be an immediate outcome of the loss of KIF20A function, although LOF of KIF20A can lead to an eventual cell fate change.

We agree that the precise mechanism of how RGS-KIF20A works to regulate cell fate specification is not yet understood from our current data. We are inclined to believe that a better understanding of this will most likely come from comprehensive functional identifications of additional intercellular bridge (ICB) resident factors that are involved in cell fate decision making.

3) I understand that there is of course variation in IUE experiments. Nevertheless, it is the responsibility of the authors to ensure that experiments are made precisely at the reported stages, or the results will carry significant doubt. Also, my point remains that 48h are definitely enough for a substantial amount of electroporated wt progenitors to differentiate and migrate to the CP. This means that scr experiments should look like Fig.2b Scr., rather than Fig.2a Scr. or Fig.4a GFP. This being the case, the difference between Scr. and shKIF20A would not appear to be so significant. In addition, these variations in the Scr. data argue that your GFP plasmids and IUE procedure may be themselves responsible and need to be thoroughly controlled and perhaps replaced. Otherwise, significant portions of the main manuscript conclusions are not clearly interpretable.

In sum, while improvements have definitely been made, these still seem insufficient to justify publication.

To minimize the effect of variations of IUE assays, whenever possible, we had paired two different plasmids for IUE in the same mouse to obtain littermate brains. Also, quantifications were focused to compare distributions of transfected cells in different radial domains of the cortex, rather than absolute numbers in each domain. In our experience, IUE-based phenotype of induced differentiation was concomitant with a strong upward shift of transfected cells from VZ into IZ and CP, comparing to a control plasmid transfected littermate brains.

To address your concern further, we have performed experiments focusing on obtaining additional data comparing scrambled shRNA with shKif20a-3UTR or scrambled shRNA with shKif20a-CDS electroporated in pairs in the same mouse. In these experiments, plasmids were electroporated at the day of E13.5 based on plug date calculation (due to variations of the exact mating time, embryos of different litters could be at varied stages, e.g. E13, E13.5 or E14), and all littermate brains were collected at the same two day duration post IUE. The Figure attached below showed some examples of patterns of littermate brains (in which we could recover embryos of paired plasmids from the same mouse).

As shown, distributions of control cells injected with scrambled shRNA varied in different litters, but within each litter, cells injected with shKif20a consistently displayed noticeably significant upshift of distribution away from the VZ into the IZ and CP regions, compared to control cells. In cases when control cells showed many cells in the CP region (e.g. Litter 1), cells of littermates injected with shKif20a displayed more significant depletion of cells in the VZ region, an effect perhaps compounded by Kif20a knockdown-induced differentiation/outward migration plus an apparent litter-specific earlier neurogenesis (more control cells moved into the CP than in other litters). Overall, knockdown of Kif20a caused marked upward shifts of transfected cells away from the VZ in the cortex. To better represent the distribution of cells, we revised Figure 2a of Scrambled shRNA, shKif20a-CDS and shKif20a-3UTR using the new data.

Most importantly, while IUE-based knockdown experiments could be very useful for getting initial insights into the function of a gene of interest (GOI), the subsequent genetic knockout would be a more conclusive and essential approach for validation of the biological relevance of the GOI. Here in this manuscript, our conclusion that KIF20A is an essential cell fate determinant regulating the balance between proliferation and differentiation in NPCs was supported by both IUE-based and genetic-based experiments.

Reviewers' Comments:

Reviewer #2:

Remarks to the Author:

The point 1 has been addressed in an acceptable way. However, the points 2a and 2b have not been.

2-a. The authors' explanation by the non-uniform timings of mating might make sense. I tried to find examples of the grainy appearance of colonies in published papers. However, I failed to find one. Can the authors show us such an example?

2-b. I appreciate that the authors performed the experiment I suggested. However, the intensity of the "Kif20a" band in the new Figure S2d is surprisingly strong, to be honest, too strong to be true. The exact details of the volumes of the lysate used as an input, the sample buffer to re-suspend the immunoprecipitates and the samples loaded on the gel have not been described. Thus, precise discussion on the quantities on the final blot that we can/should expect is impossible. However, based on the blots in Figure 2b and 2c, even at the maximum efficiency (direct IP instead of co-IP), we can probably expect only a similar level to that on the input lane. I don't understand why the authors don't show the input in the new Figure S2d while they properly show the inputs for Figure S2b and c. Disappointingly, this is a serious flaw.

Anyway, the only data detecting Kif20a in a crude lysate in this manuscript is Figure S7d. The band of Kif20a in this panel is very thin and its intensity is very low. We can't compare the intensities of two bands on separate blots. However, based on the shapes of the bands, the "Kif20a" band on the "IP:RGS3 (brain lysate)" seems to be at least 100-times stronger than the "Kif20a" band on the "WT" lane in the Figure S7d panel. The amount of protein detected by co-immunoprecipitation via a protein-protein interaction is usually lower than that precipitated directly (eg, myc-tagged protein by an anti-myc antibody). Thus, the blot on Figure S7d is extremely unusual and, thus, the annotation is difficult to believe.

The authors should repeat the western blotting by including the lysate used as an input on the same gel and blot (similarly to Figure S2b and c, with an empty lane or a marker, something guaranteed for no signal, to avoid inter-lane leakage/contamination). SDS-PAGE should be better performed with a gel of a lower percentage of polyacrylamide to maximize the separation of the bands in the range between 95~130 kDa. Then, they should confirm that 1) the mobilities of the Kif23a bands in the input and co-IP are identical and 2) the Kif23 protein amount co-immunoprecipitated by RGS3 is not more than 100% of the Kif23 protein in the input.

Reviewer #3:

Remarks to the Author:

Comments on the 2nd revised version of "KIF20A/MKLP2 regulates the division modes of neural progenitor cells during cortical development". Geng and colleagues have further answered some reviewer's questions and have added more data to clarify some remaining points, and have further tried to find mechanistic insights. Some issues should be addressed before publication, however, including some pointed out since the first round of revision.

1) Throughout the text and in some titles, the authors still repeatedly claim that neuronal differentiation is an outcome for several experiments that actually do not show this (for example in Fig2 a & b, and Fig.4a). Also, the title in p.9 refers only to isolated progenitors that are not in cortical tissue. Those experiments show an interesting depletion of GFP+ cells from the VZ/SVZ, but the cells that migrated in higher proportions past those zones have not been shown to be neurons. Mere CP localization is simply not enough. The cells in those figures could be crucial, since they are the only tissue cells in the manuscript where it is possible to directly follow the

acute effects in fate of depleting KIF20A. Either the authors modify the respective text to reflect the actual results, or they should provide IFs with neuronal nuclear markers (Cytoplasmic BIII-tub, as in Fig7b and S9a, can be inaccurate for conclusively assigning marker positivity to specific nuclei).

In the KO mice, the authors do show progenitors and a thinner CP, but there is also significantly more apoptosis at E12.5, when progenitors may still be proliferating (Fig.6b). Thus, an alternative explanation to “precocious neuronal differentiation” may be that early apoptosis could be sufficient to cause those depletions, because some early progenitor apoptosis can have strong cumulative effect in later stages.

Together, these uncertainties, in both the IUE and KO experiments, mean that a central claim of the manuscript, that KIF20A “leads to a fate transition from proliferative to differentiative divisions” remains insufficiently documented in the context of cortical tissue. For publication, either all over-interpretations should be eliminated, or the claims should be sufficiently supported.

2) Former point 3) Discrepancies in IUE experiments have been improved, except in Fig.4a “GFP”. It puzzles me that the authors insist in using control IUEs with very few GFP+ CP cells, as this clearly is not normal, nor representative of most images they themselves have provided. This should be corrected in all relevant places. Also, there is a strong band of GFP+ cells near the pial surface of the E15.5 GFP-PDZ-RGS3-IRES-ephrin-B1, which is not mentioned in the text, and may even be neurons. Please clarify.

The authors would also do well to review and describe their statistical methods in more detail, as it is not clear whether T-tests are the correct tool to compare multiple variables that are not necessarily independent of each other, for example in Fig.2.

Response to reviewers' comments

Referee #2

The point 1 has been addressed in an acceptable way. However, the points 2a and 2b have not been.

2-a. The authors' explanation by the non-uniform timings of mating might make sense. I tried to find examples of the grainy appearance of colonies in published papers. However, I failed to find one. Can the authors show us such an example?

No. The explanation was our speculation based on the nature of this experiment.

2-b. I appreciate that the authors performed the experiment I suggested. However, the intensity of the "Kif20a" band in the new Figure S2d is surprisingly strong, to be honest, too strong to be true. The exact details of the volumes of the lysate used as an input, the sample buffer to re-suspend the immunoprecipitates and the samples loaded on the gel have not been described. Thus, precise discussion on the quantities on the final blot that we can/should expect is impossible. However, based on the blots in Figure 2b and 2c, even at the maximum efficiency (direct IP instead of co-IP), we can probably expect only a similar level to that on the input lane. I don't understand why the authors don't show the input in the new Figure S2d while they properly show the inputs for Figure S2b and c. Disappointingly, this is a serious flaw.

With respect to input lanes, in transfected Hek cell co-IP experiments, input lanes were used to verify that plasmid transfection/expression went efficiently so that any potential negative IP or co-IP results could be ruled out as a result of lacking expression of the proteins of interest, such as myc-PDZ-RGS3, myc-PDZ-RGS3 Δ RGS and Flag-KIF20A in Figure S2b and S2c. Particularly in Figure S2b, we need the input lanes to validate that myc-PDZ-RGS3 (full-length protein) and myc-PDZ-RGS3 Δ RGS (deletion mutant) were expressed to a similar level, so the result of the mutant protein failing to co-IP with KIF20A was not due to its inadequate expression than that of the full-length protein, but due to deletion of the RGS domain (the KIF20A-binding domain). On the other hand, in the case of co-IP of endogenous proteins, both anti-RGS antibody and the control IgG were mixed with the same brain lysate, so direct comparison between lane #1 and lane #3 in Figure S2d was the key to observe.

Anyway, the only data detecting Kif20a in a crude lysate in this manuscript is Figure S7d. The band of Kif20a in this panel is very thin and its intensity is very low. We can't compare the intensities of two bands on separate blots. However, based on the shapes of the bands, the "Kif20a" band on the "IP:RGS3 (brain lysate)" seems to be at least 100-times stronger than the "Kif20a" band on the "WT" lane in the Figure S7d panel. The amount of protein detected by co-immunoprecipitation via a protein-protein interaction is usually lower than that precipitated directly (eg, myc-tagged protein by an anti-myc antibody). Thus, the blot on Figure S7d is extremely unusual and, thus, the annotation is difficult to believe.

With respect to signal intensities, in the experiment of Figure S7d, a portion of whole cell lysate of individual E12.5 wild-type or homozygous knockout forebrain tissue was used per lane to validate germline knockout among the littermates (older stages could yield brains with more proteins for western blot but were associated with more prominent lethality in the homozygous mutants thus not allowing simultaneous recovery of WT and Null littermate brains). On the other hand, co-IP was enriched product from multiple E15.5 wild-type forebrains. So the two experiments were not directly comparable. In addition, it is generally known that western blot is

not linear across the entire range, i.e. there is a technical threshold for protein detection associated with antibody recognition/detection. For example, two samples of the same protein, one is 20 ng/ μ l and the other is 5 ng/ μ l, if 5 ng/ μ l is somehow at the borderline of threshold of detection, the blot of band signal of the 5 ng/ μ l sample will not be one quarter intensity of that of 20 ng/ μ l sample as anticipated, it could be 10 times weaker or even barely detectable. I am therefore having difficulty in understanding this comment of using much of quantitative terms of band intensities, particularly inferring a weaker signal to other bands of different intensities.

The authors should repeat the western blotting by including the lysate used as an input on the same gel and blot (similarly to Figure S2b and c, with an empty lane or a marker, something guaranteed for no signal, to avoid inter-lane leakage/contamination). SDS-PAGE should be better performed with a gel of a lower percentage of polyacrylamide to maximize the separation of the bands in the range between 95~130 kDa. Then, they should confirm that 1) the mobilities of the Kif23a bands in the input and co-IP are identical and 2) the Kif23 protein amount co-immunoprecipitated by RGS3 is not more than 100% of the Kif23 protein in the input.

--

Data of a repeated co-IP of endogenous proteins from brain lysate were shown in the new Figure s2d. Experimental details were included in the Methods section. In this experiment, the proteins of higher molecular weights were let run more extensively during electrophoresis, in order to “**maximize the separation of the bands in the range between 95~130 kDa**” as requested (lowering the percentage of gel below the 7.5% separating gel that we used would not help in this regard). In this experiment, one major protein reactive to anti-KIF20A antibody was detected in both the input and co-IP, migrating at a size anticipated for KIF20A protein, and this protein band was absent in the lane of an IgG control.

Referee #3

Comments on the 2nd revised version of “KIF20A/MKLP2 regulates the division modes of neural progenitor cells during cortical development”. Geng and colleagues have further answered some reviewer’s questions and have added more data to clarify some remaining points, and have further tried to find mechanistic insights. Some issues should be addressed before publication, however, including some pointed out since the first round of revision.

1) Throughout the text and in some titles, the authors still repeatedly claim that neuronal differentiation is an outcome for several experiments that actually do not show this (for example in Fig2 a & b, and Fig.4a). Also, the title in p.9 refers only to isolated progenitors that are not in cortical tissue. Those experiments show an interesting depletion of GFP+ cells from the VZ/SVZ, but the cells that migrated in higher proportions past those zones have not been shown to be neurons. Mere CP localization is simply not enough. The cells in those figures could be crucial, since they are the only tissue cells in the manuscript where it is possible to directly follow the acute effects in fate of depleting KIF20A. Either the authors modify the respective text to reflect the actual results, or they should provide IFs with neuronal nuclear markers (Cytoplasmic BIII-tub, as in Fig7b and S9a, can be inaccurate for conclusively assigning marker positivity to specific nuclei). In the KO mice, the authors do show progenitors and a thinner CP, but there is also significantly more apoptosis at E12.5, when progenitors may still be proliferating (Fig.6b). Thus, an alternative explanation to “precocious neuronal differentiation” may be that early apoptosis could be sufficient to cause those depletions, because some early progenitor apoptosis can have strong cumulative effect in later stages.

Together, these uncertainties, in both the IUE and KO experiments, mean that a central claim of the manuscript, that KIF20A “leads to a fate transition from proliferative to differentiative divisions” remains insufficiently documented in the context of cortical tissue. For publication, either all over-interpretations should be eliminated, or the claims should be sufficiently supported.

- With regard to “Those experiments show an interesting depletion of GFP+ cells from the VZ/SVZ, but the cells that migrated in higher proportions past those zones have not been shown to be neurons”, we have included data showing cells migrated into the CP under expression of shKif20a were positive for nuclear neuronal marker NeuN (new Figure 2b). In addition, we have also revised Fig7b and S9a with new data (replacing beta III-tubulin staining) showing cells that moved early into the CP were positive for NeuN staining. These results are consistent with the phenotype of induced early neuronal differentiation, in which blocking KIF20A function causes cell cycle exit in the affected NPCs and subsequent outward migration of the earlier-born neuronal progeny.
- With regard to the comment of “In the KO mice, the authors do show progenitors and a thinner CP, but there is also significantly more apoptosis at E12.5, when progenitors may still be proliferating (Fig.6b). Thus, an alternative explanation to “precocious neuronal differentiation” may be that early apoptosis could be sufficient to cause those depletions, because some early progenitor apoptosis can have strong cumulative effect in later stages”, the overall numbers of caspase 3-positive or TUNEL-positive cells were very limited within the entire cortical regions in both the wild-type and knockout brains (even with an increase from the wild-types) (Figure 6a and 6b), thus the increase of apoptosis at E12.5 did not seem to be able to account for, but could be one factor contributing to, the level of the overall loss of neural progenitor cells in the Kif20a knockout, as we have discussed in the Discussion section of the manuscript.

2) Former point 3) Discrepancies in IUE experiments have been improved, except in Fig.4a “GFP”. It puzzles me that the authors insist in using control IUEs with very few GFP+ CP cells, as this clearly is not normal, nor representative of most images they themselves have provided. This should be corrected in all relevant places. Also, there is a strong band of GFP+ cells near the pial surface of the E15.5 GFP-PDZ-RGS3-IRES-ephrin-B1, which is not mentioned in the text, and may even be neurons. Please clarify.

The authors would also do well to review and describe their statistical methods in more detail, as it is not clear whether T-tests are the correct tool to compare multiple variables that are not necessarily independent of each other, for example in Fig.2.

- With regard to “there is a strong band of GFP+ cells near the pial surface of the E15.5 GFP-PDZ-RGS3-IRES-ephrin-B1, which is not mentioned in the text, and may even be neurons. Please clarify”, this was apparently within the meninges, because in some other cases, the ribbon of the green signal was clearly separated atop from the cortical regions. However, we are not clear whether these are true GFP signals or a background signal associated with the meninges.
- With regard to “The authors would also do well to review and describe their statistical methods in more detail, as it is not clear whether T-tests are the correct tool to compare multiple variables that are not necessarily independent of each other, for example in Fig.2”, we have now used separate asterisks in different colors to match the CP and VZ/SVZ regions in the graph

(Figure 2a and Figure 7c), in addition to the description of the asterisks in the Figure Legends.

- With regard to “**Former point 3) Discrepancies in IUE experiments have been improved, except in Fig.4a “GFP”**”, I guess that I miscommunicated on this issue previously and I should clarify it. In Figure 2a, the control panel was a scrambled shRNA together with GFP (the plasmid contains a shRNA expression cassette and a GFP expression cassette in the same vector). With such controls (GFP + scrambled shRNA), there were some variations in distributions of cells in the CP, as we presented and discussed in the data included in our previous response. We speculate that this variation was most likely caused by a general effect associated with expression of hairpin RNAs (see discussion at the end).

In Figure 4a, the GFP control panel was GFP alone (without the presence of any hairpin RNAs) for comparing with the panel showing co-expression of RGS3 and Ephrin-B1 proteins. In the case of GFP alone, our data have been much more consistent and that in most cases not many cells had migrated into the CP two days after electroporation (E13.5 IUE and E15.5 for analysis). Some additional images of GFP (or mCherry) alone at E15.5 from different batches (brains) of our previous and ongoing experiments were shown in the Figure below. The GFP image in Figure 4a seemed to be a fairly representative picture reflecting this control condition.

Therefore, it appeared that variations of GFP controls in IUE-based RNAi experiments are associated with the presence of hairpin RNAs. To extend from this speculation, I like to share some more of our experience and thoughts on the possible cause of this issue based on our unpublished data. Since the initial use of 19mer short hairpin RNAs in RNAi, longer shRNAs such as 21mer and 27mer were shown to have stronger knockdown effect than

19mer shRNAs, thus many researchers (and commercially available shRNAs) switched to use longer shRNAs. However, it was also found that shRNAs were associated with activation of interferon responses and that the interferon response was positively correlated with the length of shRNAs, with 19mer shRNAs showing the least effect. In our IUE-based assays, we had also tested longer shRNAs (e.g. 21mers) in hope to obtain better knockdown effects of target genes, however, we found that longer shRNAs, such as 21mer scrambled shRNA controls, could often mobilize transfected cells to move into the IZ and CP, which would obscure potential analysis of neuronal differentiation by knockdown of a target gene. We speculated that this was most likely due to activation of interferon responses by such long shRNAs, compounded with the reported effect of interferons in promoting neural stem/progenitor cell differentiation (thus causing affected cells to show strong up-shift in cell distribution into the IZ and CP). In the case of 19mer shRNAs used in most of our experiments such as in Figure 2a, the observed variations of cell distributions in IUE controls were likely also originated from varied but low interferon responses in different litters or brains, giving rise to mild up-shift of cell distributions in some brains. However, such variations did not deviate significantly from GFP alone controls (the baseline of distribution of GFP+ cells in the IUE assays, as shown in Figure 4a and the attached Figure above) and did not obscure analyses of neuronal differentiation phenotypes in our hands, as we had presented and discussed in the data included in our previous response.